# Quantum scale organic semiconductors for SERS detection of DNA methylation and gene expression

Swarna Ganesh[1,2,3], Krishnan Venkatakrishnan [2,3,4 ✉] & Bo Tan [3,4,5]

Cancer stem cells (CSC) can be identified by modifications in their genomic DNA. Here, we report a concept of precisely shrinking an organic semiconductor surface-enhanced Raman scattering (SERS) probe to quantum size, for investigating the epigenetic profile of CSC. The probe is used for tag-free genomic DNA detection, an approach towards the advancement of single-molecule DNA detection. The sensor detected structural, molecular and gene expression aberrations of genomic DNA in femtomolar concentration simultaneously in a single test. In addition to pointing out the divergences in genomic DNA of cancerous and non-cancerous cells, the quantum scale organic semiconductor was able to trace the expression of two genes which are frequently used as CSC markers. The quantum scale organic semiconductor holds the potential to be a new tool for label-free, ultra-sensitive multiplexed genomic analysis.

[1] Institute for Biomedical Engineering, Science and Technology (I BEST), Partnership between Ryerson University and St. Michael's Hospital, Toronto, ON M5B 1W8, Canada. [2] Department of Mechanical and Industrial Engineering, Ultrashort Laser Nanomanufacturing Research Facility, Ryerson University, 350 Victoria Street, Toronto, ON M5B 2K3, Canada. [3] Department of Mechanical and Industrial Engineering, Nano Bio Interface facility, Ryerson University, 350 Victoria Street, Toronto, ON M5B 2K3, Canada. [4] Keenan Research Center, St. Michael's Hospital, 209 Victoria Street, Toronto, ON M5B 1T8, Canada. [5] Nanocharacterization Laboratory, Department of Aerospace Engineering, Ryerson University, 350 Victoria Street, Toronto, ON M5B 2K3, Canada. ✉email: venkat@ryerson.ca

Surface-enhanced Raman scattering (SERS) analysis is a newcomer in cancer detection. It provides real-time molecular information at high resolution without the use of labels. However, it suffers from weak signal and limited penetration depth, typical for optical based techniques. However, current research on SERS have been largely dominated by plasmonic materials. Despite providing a high enhancement efficiency, the conventional probes suffer from inherent disadvantages such as poor stability, poor biocompatibility, complex synthesis process along with high cost of fabrication[1]. Hence, there is always a constant search for new class of materials to replace metallic probes.

Organic semiconductors have demonstrated the potential to replace metallic probes as a reliable SERS sensor. Although, SERS offers the necessary ultra-sensitive, multiplex detection attributes to get a holistic picture of epigenetic landscape, the interaction of existing SERS probes with DNA greatly alters the native structure of DNA leading to inappropriate diagnosis. To date, there exists no single method to detect the epigenetic component of cancer.

SERS with organic semiconductors presents distinctive advantages, including structural adaptability, tunable charge mobility, low effective carrier mass[2]. Recently reported organic semiconductor sensors are thin films[3,4]. However, organic semiconductors rely primarily on charge transfer mechanism for SERS enhancement which hinders the sensitivity to a great extent. Due to these limitations, the use of organic semiconductors as SERS sensor is scarce, although organic semiconductors offer many attractive advantages than its inorganic counterparts in terms of biocompatibility. In addition, carbon-based organic semiconductor probes are chemically stable in the cell culture medium[5] and remain inactive in the cellular microenvironment[6]. These properties make organic semiconductors uniquely qualified to be used as a biological sensor.

In the last decade, much research attention has been focused on cancer stem cells. Studies have revealed that the survival of CSC is a primary obstacle to effectual cancer therapy[7,8]. CSC is characterized by multi-faceted alterations in multiple molecular systems. Various molecular analysis methods have been explored in recent years in searching for a multi-parametric classification/ analysis tool to determine the molecular aberrations of CSC[9]. The genomic analysis provides not only the complete information about the structural and molecular changes of DNA but also the changes in gene expression. The analysis of epigenetic changes in DNA may help to identify new druggable targets for the treatment of cancer[10]. Investigating the epigenetic landscape of genomic DNA involves the analysis of molecular (methylation), structural (base composition) and functional (gene expression) states of the DNA. Traditional methods to sense molecular changes in genomic DNA include but not limited to pyrosequencing and mass spectrometry[11]. These methods involve extensive chemical modification of DNA, which alters the native structure of DNA thereby rendering the results unreliable[11].

Single-molecule genomic DNA analysis techniques have been proposed[12]. Although these single-molecule techniques offer certain advantages compared to traditional techniques, real-time application is severely hindered by their high error rate, short read lengths, and inability to analyse complex genomes[13]. The research of single-molecule genomic DNA analysis is still at its infant stage. Conventional methods of gene expression analysis using PCR is semi-quantitative and present numerous technical difficulties[14,15] and the ability to detect multiple genes simultaneously is severely limited due to the need for multiple templates and the need to amplify the gene to be suitable for detection by existing methods.

From the above discussion we can see that to get accurate information of genomic contribution to the origin of cancer, a detection method that is non-invasive, label-free, high sensitivity, capable of capturing multiplex gene information[16,17] is desired. To get a holistic picture, it is desirable to collect information of all components simultaneously. None of the existing method provides simultaneous detection of multiple genes.

In this article we report a tag free quantum organic semiconductor (QOS) sensor aimed to detect cancer stem cell phenotype by utilizing single-molecule genomic DNA. The genomic DNA holds key information about the structural, molecular and genetic modifications, which may be effective markers for cancer stem cells. The ultrashort pulsed laser processing in the presence of nitrogen gas enabled the shrinking of organic semiconductor to quantum scale, which resulted in increased charge carrier mobility essential for efficient charge transfer necessary for SERS enhancement. Detection of CV and R6G demonstrated a label-free single-molecule detection (femtomole) sensitivity with an enhancement factor of $10^{12}$, which is 9 orders higher than reported value from organic semiconductors. These results motivated us to test QOS for the detection of epigenetic markers of cancer stem cells. The genomic DNA isolated from various cellular models were used to determine the structural and molecular changes using SERS. The practical applicability of tag free QOS sensor was validated using genomic DNA isolated from four different cell lines, namely fibroblast cells (NIH3T3), breast cancer (MDA-MB 231), pancreatic cancer (AsPc-1), and lung cancer (H69-AR). Base composition of DNA and methylation markers are collected with one single detection. By performing multivariate statistical analysis, we could determine the molecular differences between the genomic DNA of cancerous and non-cancerous cells. Next, QOS was studied for gene expression analysis. The expression of two genes, commonly used as markers of CSC, were analyzed. The ability of QOS to study the epigenetics of cancer stem cells, has opened the possibility of extending the utilization for non-invasive, personalized cancer diagnosis. This study opens new possibilities with organic sensors. It holds potential not only for disease biomarkers detections but also other applications requiring ultra-sensitive detection, such as rapid sensing of environmental contaminants, hazardous chemicals, and explosives.

## Results

**Synthesis of quantum organic semiconductor probes**. The quantum-sized organic semiconductor probes were synthesized using a femtosecond laser pulse irradiate on a graphite substrate in the presence of low-pressure nitrogen gas as shown in Fig. 1a, b. When a carbon plasma is induced on the graphite surface, it reaches a high temperature and pressure. The expansion of plasma happens in the direction perpendicular to the surface of graphite. The rapid expansion of plasma plume is condensed by the presence of low-pressure gas in the background. The surrounding gas plays a prominent role in defining the dynamics and contents of the laser ablated plasma[18]. In the presence of low-pressure nitrogen, the velocity of carbon plasma decreases resulting in reduction in ionic collision frequency, which eventually increases plasma density. Further, the integration of nitrogen gas to the plume environment helps to condense the molecular species, thereby increasing the recombination rate.

The fragmented materials expelled from the laser interaction zone is primarily make up of charged ions, atoms, and cluster of particles with various atomization states based on laser fluence[19]. The molecular/ionic species in the carbon plasma at high-laser fluence is dominated by $C^+$, $C_2$, $CN$, $N_2^+$, neutral species of C, N. The chemical composition of the quantum scale particles formed by the condensation of plume can be attributed to the impact of fast electrons leading to ionization and excitation of nitrogen gas

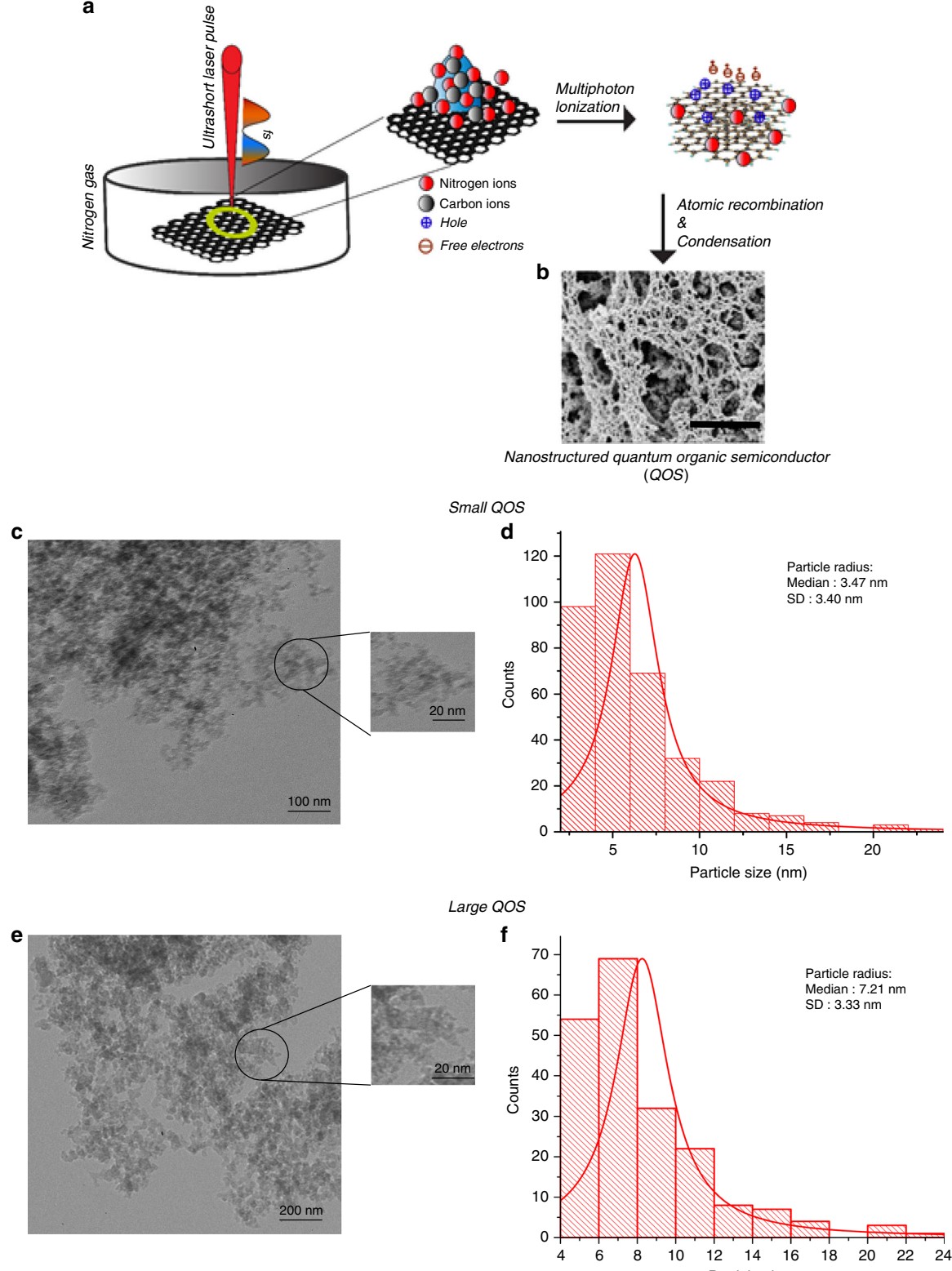

**Fig. 1 Synthesis of Quantum Organic Semiconductor Probes. a** Schematic of shrinking organic semiconductor using a femtosecond laser pulse under low-pressure nitrogen atmosphere, **b** top-view scanning electron microscope image showing the interconnected quantum organic semiconductor probes, **c**, **e** morphology of Small QOS and Large QOS respectively, **d**, **f** Size distribution of Small QOS and Large QOS respectively showing the normal distribution of probes.

surrounding the plume. Studies have proven that CN molecules are formed by the high-speed collision between $C_2$ and $N_2$ molecules in the gas phase. The saturation of CN molecules causes collision with $C_2$ molecules resulting in the expansion of carbon plasma.

The $C_2$ molecules in the laser ablated plume is a result of a combination of various mechanisms, including recombination of carbon atoms and ions, fragmentation/dissociation of carbon clusters formed during the initial states of carbon plume.

Research based on plasma emission spectroscopy revealed that the most probable mechanism for the formation of $C_2$ species is by recombination of carbon atoms in the plume, with assistance from a background gas. The process can be described using the following equation:

$$C + C + N_2 \rightarrow C_2 + N \quad (1)$$

The formation of CN species can be explained by the reaction:

$$C_2^* + N_2 \rightarrow 2CN^* \quad (2)$$

The "∗" indicates the excited state of molecular species. The formation rate of each of the molecular species is determined by the individual bond energies[20]. Thus, the determination of molecular species, helps in regulating the functional groups present in the organic semiconductor probes. Further, the incorporation of heteroatoms, such as nitrogen, in the basal graphene architecture is instrumental in tailoring the semiconductor properties of the probe.

The method used in this work to shrink organic semiconductors have overcome the above-mentioned limitations, thus aiding in precisely manipulating both physical (size) and chemical (functional groups) properties of the organic semiconductors, thus creating a efficient, uniform SERS substrate.

Figure 1 presents the morphological characterization of the laser synthesized probe. A transmission electron microscopy image represented in Fig. 1c, e shows agglomerated particles of QOS with a median particle size distribution (Fig. 1d) of 3.4 nm for the small QOS and 7.2 nm for large QOS (Fig. 1f). During the process of multiphoton ionization, the formation of particles in quantum scale regime always occurs independent of the wavelength. The varying size distribution of the organic semiconductor probes indicates the formation of probes occurred in a non-equilibrium state.

Studies have revealed that the semiconductor properties of graphene based materials can be manipulated by introducing heteroatoms, such as nitrogen[21]. The incorporation of nitrogen atoms in graphene leads to various bonding configurations including pyridinic N (C–N bonds in the aromatic rings), pyrrolic N (two C–N bonds in pentagonal aromatic ring), quaternary N (three sp2 C–N bonds). The incorporation of nitrogen contributes one p electron to the π system[22]. In the presence of oxygen, the nitrogen atom bonds with two carbon atoms and one oxygen atom.

Chemical characterization of the probe is done with XPS and Raman spectroscopy. The results are given in Supplementary Figs. 1 and 2, respectively. The detailed explanation is provided in Supplementary Note 1 and Supplementary Note 2. The XPS analysis (Supplementary Fig. 1b) showed the presence of COOH bonds only in the large QOS and not in the small QOS. Further, the presence of nitrogen atoms adjacent to carbon modifies the charge distribution of the system, resulting in the creation of "activation zone". The activated zone enhances adsorption of molecules, thereby inducing efficient charge transfer. The optical characterization of QOS is presented in Supplementary Fig. 3. The detailed explanation is provided in Supplementary Note 3. Based on the band gap, XPS valence band spectra, the band structure of QOS was elucidated and shown in Supplementary Fig. 3b. It can be understood from the position of valence band and conduction band that it is possible to find potential charge-trapping sites, which can ascertain the mechanism of SERS enhancement.

**Molecule sensing**. The SERS characteristics of the QOS were investigated using Crystal Violet (CV), Rhodamine 6G (R6G). Figure 2 shows the SERS spectra of CV, and R6G, respectively, at

an excitation wavelength of 785 nm. It can be observed from Fig. 2 that Raman spectra of analyte molecules on bare graphite substrate show extremely weak peaks, confirming no significant Raman enhancement on the graphite substrate. Interestingly, the small QOS network exhibits strong differences in SERS enhancement. These results validate that the QOS can function as a SERS active architecture. The characteristic SERS peaks at 1190 cm$^{-1}$ for R6G and 990 cm$^{-1}$ for CV, corresponding to in-plane C–H bending vibration, were used to calculate the enhancement factor. The detailed enhancement factor calculation is provided in the methods section of the paper. The resulting enhancement factor is $1.106 \times 10^{12}$ for crystal violet and $4.56 \times 10^{12}$ for rhodamine 6G for picomolar concentration.

Currently, the enhancement factor obtained using bare organic semiconductor thin films is $10^3$ and an enhancement factor of $10^{10}$ with the addition of a plasmonic (gold) layer[4]. The use of thin film organic semiconductor material relies solely on charge transfer and molecular aggregation mechanism for SERS enhancement. In the case of QOS used in this study, the achieved high enhancement factor for bare organic semiconductor is in the order of $10^{12}$. The enhancement factors for different concentration of analyte (CV and R6G) is presented in Supplementary Table 1. The enhancement factor obtained using QOS show similar values as exhibited by conventional SERS substrates like gold and silver. Furthermore, both the analyte molecules (CV, R6G) exhibited similar enhancement factor values indicating the dependence of SERS enhancement mechanism is dependent on multiple enhancement mechanism. This can be attributed to two factors. First, the high surface area for molecular adsorption. Second, the high charge carrier mobility in the organic semiconductor along with the quantum size has resulted in efficient charge transfer between analyte molecule and QOS.

The presence of high surface area for molecular adsorption has enabled the detection of molecules in an ultra-low concentration. Supplementary Fig. 4a, b shows the limit of detection of CV and Supplementary Fig. 4c, d shows the limit of detection of R6G. It can be observed from Supplementary Fig. 4a, c when the concentration of analyte decreases, the SERS intensity decreases exhibiting a linear dependence over a wide range of concentration with a R value of 0.9895 and 0.9497 for CV and R6G, respectively. It can be determined from Supplementary Fig. 4a, b that there exists a linear relationship between SERS intensity and analyte concentration, demonstrating the detection at single-molecule level. The SERS spectra of $10^{-6}$ M and $10^{-9}$ M concentration of CV and R6G is presented in Supplementary Fig. 5. It should be noted that the characteristic peaks of CV and R6G selectively enhanced at higher concentrations. This could be attributed to various factors, including molecular orientation, reduction in overall fluorescence background, particle aggregation at higher concentration leading to lower intensity of certain molecular vibrations[23].

All the SERS spectra shows a repeatable peak positions and intensities with a relative standard deviation <1.6 for the characteristic peaks of CV and R6G. In addition, for the concentration of $10^{-15}$ M the calculated signal to noise ratio for the peak 990 cm$^{-1}$ is 12 and for the peak 1190 cm$^{-1}$ is 19. This confirms that QOS is a precise substrate for detection of molecules in the sub-femtomolar range.

**SERS reproducibility analysis**. In the traditional metal-based SERS system, repeatability and reproducibility of SERS signals are the major bottlenecks. Hence, we tested the reproducibility of the obtained SERS signals by measuring the SERS spectra at five different points. The obtained spectra are shown in Fig. 3a, b for CV and R6G, respectively. It can be observed from Fig. 3a

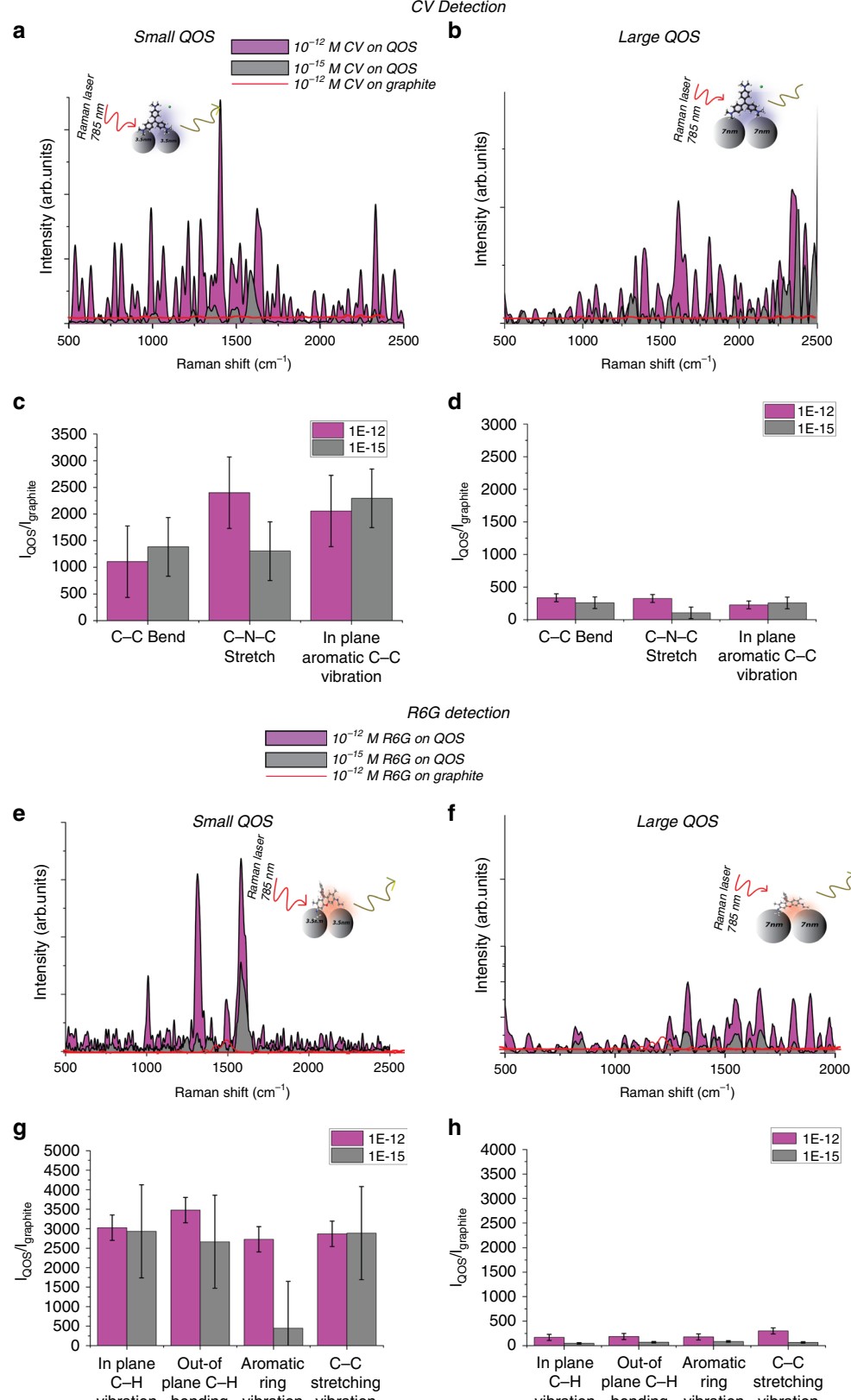

**Fig. 2 Single-molecule analyte detection by QOS.** Single-molecule detection of Crystal Violet on **a** small QOS **b** Large QOS detection of CV. **c**, **d** Enhancement efficiency of molecular bonds of as a function of $I_{QOS}/I_{graphite}$ on small QOS and large QOS, respectively. Single-molecule detection of R6G with **e** small QOS and with **f** large QOS. **g**, **h** Enhancement efficiency of molecular bonds of as a function of $I_{QOS}/I_{graphite}$ on small QOS and large QOS, respectively.

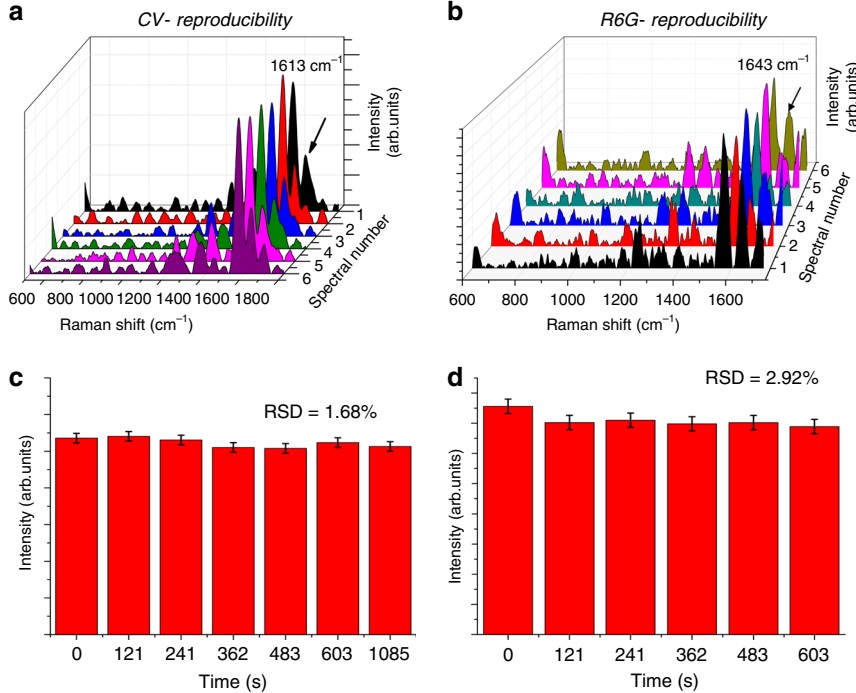

**Fig. 3 Reproducibility of SERS signals by QOS.** Reproducibility analysis of SERS signals of **a**, **b** CV, R6G on QOS, respectively, on QOS, **c** intensities of characteristic CV peak at 1613 cm$^{-1}$ shown in spectra **a** error bars: SD, **d** intensities of characteristic R6G peak at 1643 cm$^{-1}$ shown in spectra **b** error bars: SD.

showing consistent reproducible characteristic SERS peak of CV at 1613 cm$^{-1}$. The corresponding peak intensity shown in Fig. 3c at five different points show a little difference with a relative standard deviation of 1.68%, thus confirming excellent reproducibility. The similar results for R6G is shown in Fig. 3b. The spectra showcases a consistent peak at 1643 cm$^{-1}$ and corresponding uniform peak intensity with an RSD of 2.92% is shown in Fig. 3d. The presence of uniform intensity with a low RSD value indicates the reliability of nanostructured QOS as a SERS substrate.

The 3D architecture of QOS probes provides a high surface area for adsorption for the analyte molecule. Further, the presence of nitrogen in the graphene lattice helps reduce the surface energy, thus facilitating the efficient molecular adsorption. The presence of aromatic rings in the analyte molecule initiates a strong π–π interaction, thus enabling efficient molecular adsorption. Additionally, the presence of oxygen functional groups on the surface of QOS probes promotes an efficient charge transfer between the analyte molecule and QOS probes. The electronegative oxygen functional groups in the 3D architecture creates a local electric field on the analyte molecules, thus mimicking a hotspot formation upon laser excitation[24]. The XPS results presented in Fig. S1 indicates the presence of oxygen functional groups in the QOS. Recent literature has confirmed that oxygen functional groups tends to be on the surface of quantum dots. In addition, the oxygen functional groups often form a cluster like islands on the surface, which acts like hotspots. Also, when the analyte is adsorbed on the surface of QOS, the oxygenated functional group along with the local defects leads to a highly enhanced Raman signal[25]. The presence of nitrogen atoms provides additional electrons which considerably modifies the electronic properties of QOS[26,27]. The other factor that contributed to high enhancement comes from high electron mobility and the quantum size effect, which is discussed in detail in the Supplementary Fig. 6. The detailed explanation is provided in Supplementary Notes 4 and 5.

**Single-molecule DNA analysis.** The unprecedented sensitivity obtained from molecular detection motivated us to extend our experiment to epigenetic analysis of cancer cells. Four different cell lines, fibroblast cells (NIH3T3), breast cancer (MDA-MB 231), pancreatic cancer (AsPc-1), and lung cancer (H69-AR) were tested. Each cell line was probed with two separated scanning, one used to obtain Raman spectra of the genomic DNA and the other one to trace the gene expression. The obtained Raman spectrum profile contained the complete information of base composition and methylation profile. Cancer stem cell markers OCT4 and SOX2 were tracked quantitatively.

Genomic DNA was isolated according to manufacturer's protocol from four types of cellular models namely, fibroblast, breast cancer, pancreatic cancer, and lung cancer. In a standard SERS experiment, 10 μl of isolated DNA sample was placed on the nanostructured QOS. The addition of DNA to the nanostructured surface causes electrostatic interaction mediated by the negatively charged phosphate backbone and the surface of QOS. On illumination, with a 785 nm laser, the DNA-QOS molecular system yields an intense spectrum of DNA, as illustrated in Fig. 4a. In Fig. 4, we could identify the SERS peaks of DNA consistent with reported literature[28–32]. We also observe the presence of narrow and intense spectral features in the region 1000–1650 cm$^{-1}$. This spectral feature is mainly due to the interaction of bases with QOS as a result of in-plane vibrations of nucleobases[33]. The $PO_2$ peak at 1089 cm$^{-1}$ was identified as the internal standard, which helps to remove the fluctuations of absolute intensity measurements[34]. The $PO_2$ band is taken as internal standard because it is sparsely affected by DNA structural damages[35]. Further, the ratio of peak intensity ratio between the nucleotides and $PO_2$ remains constant and does not change with base composition. Hence, the ratio is considered as a marker for quantifying the base composition of genomic DNA.

The genomic DNA of the different cell lines show different peak intensity, which could be attributed to difference in base composition and difference in concentrations of the isolated

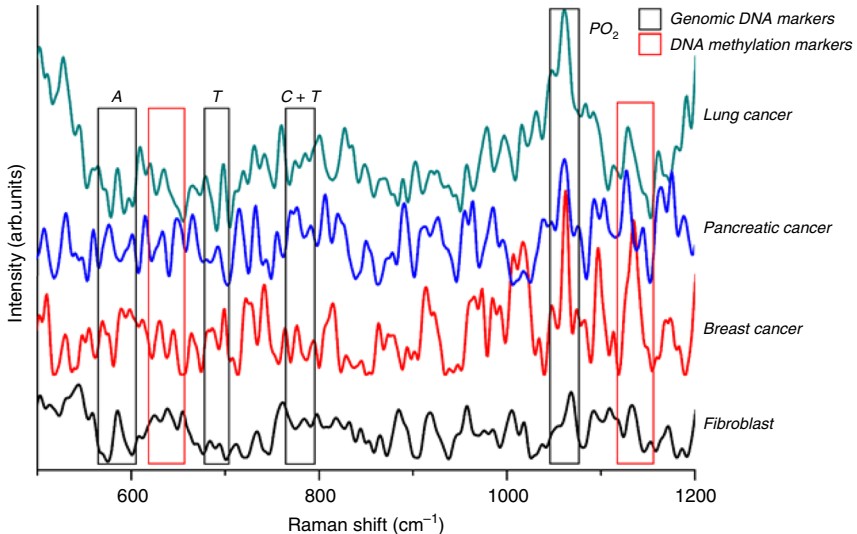

**Fig. 4 Single-molecule analysis of genomic DNA to determine base composition and DNA methylation from a single SERS measurement.** SERS spectra of genomic DNA showcasing the markers for DNA base composition analysis and DNA methylation analysis.

genomic DNA. The base composition analysis of the genomic DNA of the cellular models was performed with the peak intensities of 684 and 733 $cm^{-1}$, assigned to ring breathing mode of guanine (G) and adenine (A), respectively. The intense peak at 792 $cm^{-1}$ is a combination of ring breathing modes of cytosine (C) and thymine (T)[33]. These peaks' intensity relative to that of $PO_2$ is plotted in Fig. 5b. Each DNA sample showed the characteristic DNA peaks, but to understand the molecular differences between cancerous and non-cancerous DNA we employed PCA to classify the differences in the spectra to obtain critical information and to exclude overlapping information. Principal component analysis (PCA) is a multivariate statistical analysis tool to highlight the dissimilarity and show the patterns in a dataset. PCA of a SERS spectrum is usually carried out by reducing the dimensionality of the data making it easy to explore. In Fig. 5c, d using Principal component scores PC1 shows a 92.53% variation for cancer cell DNA and 4.47% variation for fibroblast DNA. The loading data of PC1 presented in Fig. 5c shows the dominant peaks that can help in differentiating cancer cell DNA from fibroblast DNA.

The positive peaks are 730 $cm^{-1}$ (adenine ring breathing) and 1486 $cm^{-1}$ (C=N vibration of (G + A)) and the negative peaks are 639 $cm^{-1}$ (G ring breathing), 1723 (C=O vibration of mainly G residue). These peaks characterize the differences between normal and cancerous DNA. The PC1 and PC2 cluster analysis is shown in Fig. 5d. Each data set is clustered with maximum covariance in 90% confidence ellipses. We can clearly see that the fibroblast DNA form a separate cluster from the cancer DNA. Also, the breast cancer DNA and lung cancer DNA form the positive and negative extremes of PC biplot shown in Fig. 5d, which leads to an interpretation that both samples are negatively correlated. Furthermore, the position of the clusters reveal that it is possible to differentiate cancerous DNA from normal DNA using SERS. Thus, based on the spectra and the base composition ratio, we have successfully demonstrated the use of QOS as a tag-free genomic DNA sensor with a single base sensitivity.

Methylation of genomic DNA is one of the prominent mechanisms by which cancer cells can modify downstream signaling pathways to obtain cellular senescence and the potential to invade the surrounding healthy tissues[17]. It is a significant epigenetic modifications due to addition of methyl functional group at the 5th position of cytosine and 6th position in

adenine[17,31,36]. Thus, exploring the methylation status of genomic DNA may enable us to better understand cancer[17]. Owing to the capability of QOS to sense and study the base composition of DNA, the QOS was employed to investigate the structural changes in DNA bases caused due to methylation, and results is presented in Fig. 6 SERS spectra to determine methylation of genomic DNA was obtained by placing 10 μl of DNA solution on QOS. The SERS spectra reported is statistically averaged, considering the presence of continuous Brownian motion in the scattering volume. Hence, the SERS spectra is contemplated as a true representation of the composition of the probed molecule[37]. The peaks considered for methylation analysis are consistent with the peaks reported in literature[31–33]. The direct analysis of DNA methylation using SERS happens through two independent processes involving DNA adsorption to the surface of QOS, through non-specific electrostatic interaction and base specific binding with QOS.

Among various features of DNA, we observed two distinct peaks at 733 and 760 $cm^{-1}$ (as shown in Fig. 6a, corresponding to adenine ring and cytosine ring breathing mode, respectively. The variation of intensities of these two peaks and the peak shift can be attributed to structural modification due to methylation. As given in Fig. 6c, at spectral marker of 733 $cm^{-1}$ all three cancerous cells undergo blue shift and marked decrease in intensity compared to that of Fibroblast cells. These variations in peaks indicated an addition of methyl group to adenine residue at N6 position. It can be interpreted from the spectra that methylation of adenine suffers a dramatic increase in the lung cancer and in pancreatic cancer samples.

The primary spectral marker for methylation of cytosine residue the cytosine ring breathing mode (760 $cm^{-1}$) is red-shifted for all three cancerous cells, as shown in Fig. 6d. In addition, we observed the appearance of a unique spectral feature at 740 $cm^{-1}$. It can be noted that fibroblast DNA shows a single peak at 760 $cm^{-1}$ without any evidence of peak splitting or peak shift. The cancerous cell DNA shows a red shift accompanied with the presence of peak splitting. The global hypermethylation of cancer cell DNA can also be observed from signature peak intensity as given in Fig. 5b. The normalized peak intensities of other SERS markers of methylation including peak intensities at 837 $cm^{-1}$ (C–C stretch) and 1120 $cm^{-1}$ (Adenine deoxyribose) are significantly higher than that of fibroblast (Fig. 6b). These data show that the ratio of methylation is considerably higher in

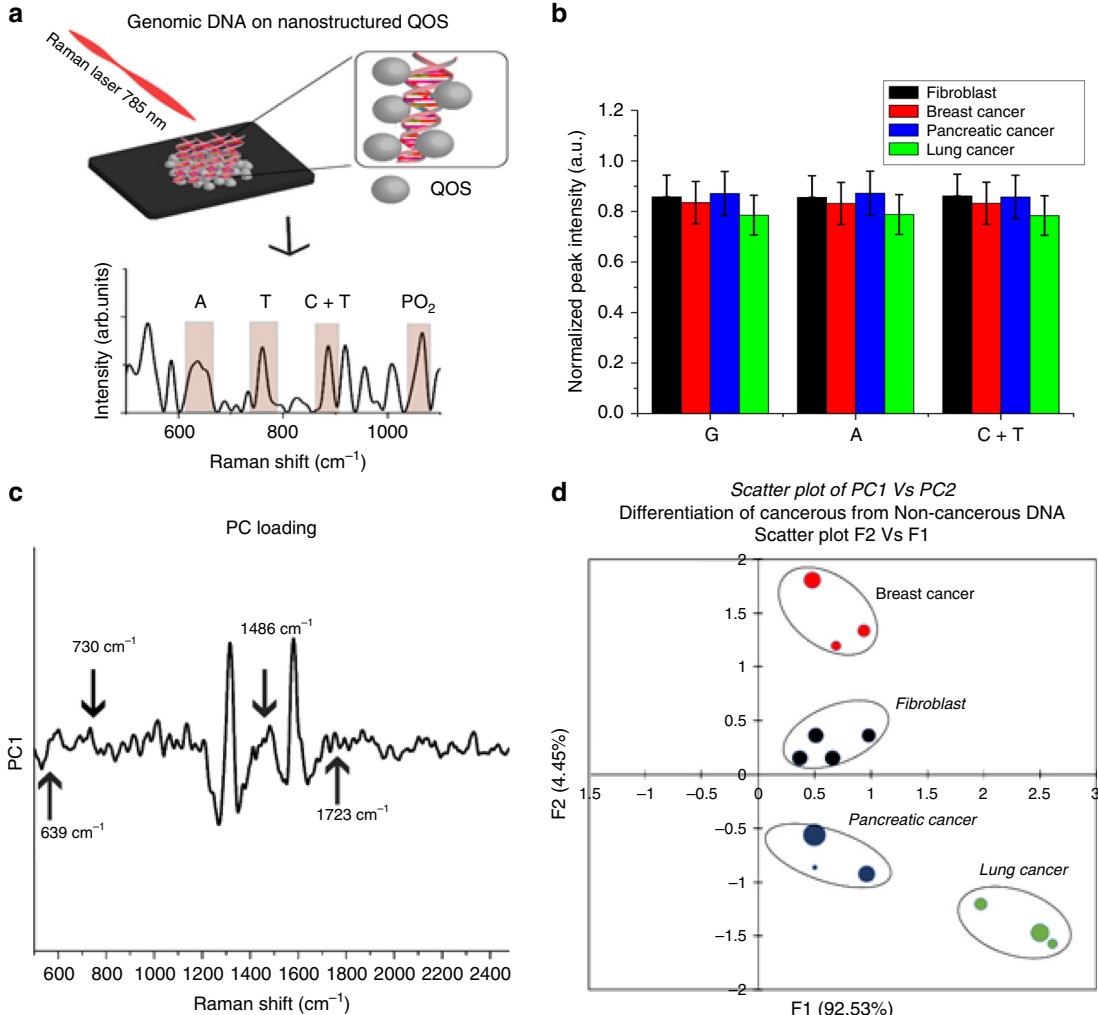

**Fig. 5 Base composition analyses of genomic DNA. a** Schematic representation of base composition analyses using SERS. **b** Normalized SERS peak intensity for various nucleic acid bases in genomic DNA of fibroblast, Breast Cancer, Pancreatic Cancer and Lung Cancer cells, **c** PC loading to determine the differences in the genomic DNA of the cellular models aiding in cancer detection **d** PCA analysis of genomic DNA derived from fibroblast, breast cancer, pancreatic cancer, and lung cancer cells.

cancer cells in comparison to fibroblast cells. DNA isolated from lung cancer cells exhibit highest degree of global hypermethylation. The applicability of QOS for real time DNA methylation detection is validated and presented in Fig. 7.

The ability of QOS to detect molecular changes in DNA further enabled us to investigate the ability to quantify global DNA methylation percentage. Figure 7a showcase the SERS spectra of 5mC DNA with different methylation percentages. It can be inferred from Fig. 7b that as the methylation percentage increases, there is a red shift in peak position accompanied with a considerable increase in peak intensity. The standard curve shown in Fig. 7c shows a linear correlation between peak intensity and methylation percentage. Hence, based on the linear fit obtained, the quantification of DNA methylation is presented in Fig. 7d. Based on the linear fit of standard curve, the methylation percentage in sample DNA was calculated with the equation: $y = 1572 + 154661 \times x$. The results (Fig. 7d) show that SERS based DNA methylation detection was as good as conventional detection systems. It should also be noted that with the QOS-based SERS sensor the detection time is greatly reduced with an improved sensitivity.

To validate proposed method of SERS to detect and quantify methylation levels, we performed the colorimetric global DNA methylation kit obtained from ABCAM (AB233486). The assay was performed according to the manufacturer's protocol and the results are presented in Fig. 7e, f. The standard curve for the colorimetric assay is presented in Fig. 7e. The methylation percentage obtained from the colorimetric assay for the DNA sample are presented in Fig. 7f.

The comparison between the DNA methylation percentage obtained from QOS-based SERS sensor demonstrated in this work and the colorimetric assay is presented in Supplementary Table 2. These results clearly show the ability of SERS based method to quantify global DNA hypermethylation levels.

Recent studies have shown that global hypermethylation is one of the epigenetic markers for identification of cancer stem cells[38,39]. DNA hypermethylation is an effective barrier for cellular reprogramming and an increased degree of global hypermethylation often leads to alteration of cellular regulatory events[40]. In addition, hypermethylation also leads to significant changes in transcription, which is regulated by gene expression[41].

To achieve tag-free gene expression analysis, QOS was employed to analyze the expression of markers for cancer stem cells, OCT-4 and SOX-2. OCT-4 is a transcription factor belonging to POU-domain transcription factors. The expression of OCT-4 is critical to determine the self-renewal potential of

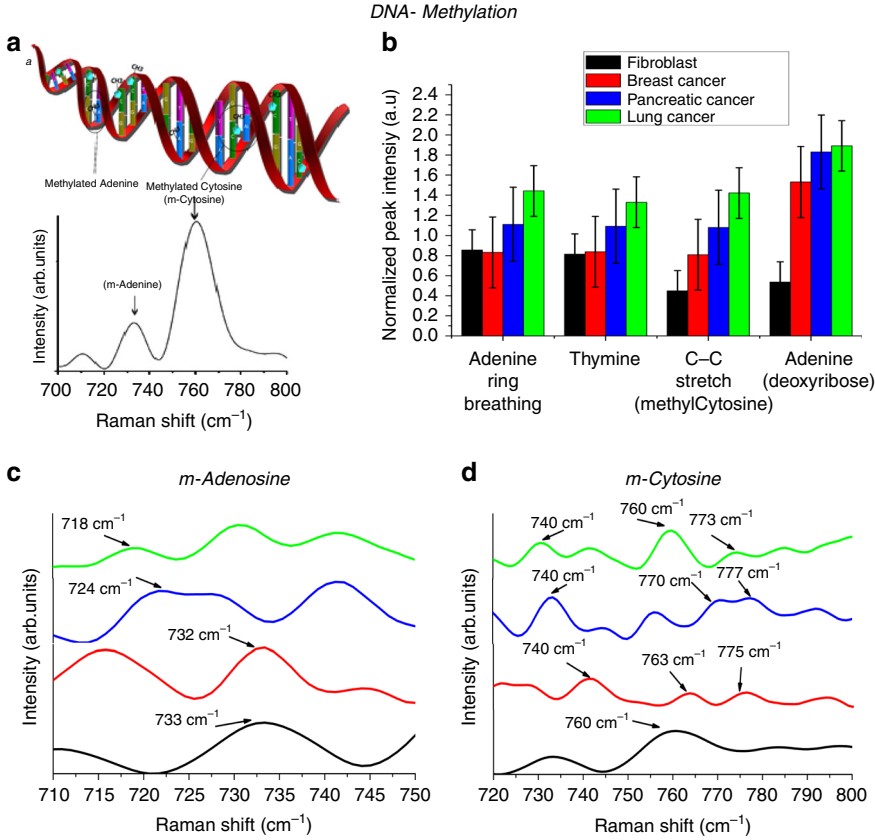

**Fig. 6 Methylation analyses of genomic DNA. a** Schematic representation of DNA methylation analysis using QOS. **b** DNA methylation markers to determine global hypermethylation in the genomic DNA isolated from fibroblast, Breast cancer, pancreatic cancer, lung cancer cells, detailed SERS spectra of the region 700–800 cm$^{-1}$ providing the markers for **c** methyl–cytosine, **d** methyl–adenine.

cells[42]. The gene SOX-2 is associated with regulation of cell cycle and cell growth[43]. Recent studies have reported the clinical implications associated with the over expression of OCT4, SOX-2. In addition, both OCT4 and SOX2 interact synergistically to control the differentiation of cells[43]. Thus, the expression of OCT4 and SOX2 were chosen as descriptive markers to detect gene expression. To standardize the process of gene expression detection and to quantify relative gene expression, we have utilized GAPDH as a housekeeping gene to serve as an internal control. The expression levels of a housekeeping gene will remain the same irrespective of the cellular origin. In each experiment, the GAPDH gene was amplified as an internal standard[42].

The detection strategy for gene expression by SERS is to perform hybridization through PCR process. PCR-assisted hybridization was then followed by SERS analysis of the hybridized sample to assess the expression levels of the corresponding genes followed by validation using conventional method of PCR amplification. The SERS spectra for gene expression analysis was obtained by placing 10 μl of the hybridized sample on QOS.

The spectral features obtained, given in Fig. 8a, c, e were consistent with the DNA peak assignments reported in literature[14,44]. The difference spectra of the respective primers were analyzed to determine the peaks distinctive peaks for the respective gene. Specifically, the three important peaks for 783, 1291, and1445 cm$^{-1}$ was used for OCT4 expression, 960, 1449, and 1910 cm$^{-1}$ for SOX2 expression analysis. The expression ratio of OCT4 and SOX2 was normalized using PO2 peak at 1087 cm$^{-1}$[35], and plotted in Fig. 8b, d, f for GAPDH, OCT-4, SOX-2 respectively. They show the relative expression of both OCT4 and SOX2 is higher in lung cancer cells.

The expression of GAPDH based on SERS intensity ratio is displayed in Fig. 8b. The results confirm the similar expression level of GAPDH on all the cell lines. The relative standard deviation of normalized peak intensity is 1.5%, 1.15%, and 0.5% for the peaks corresponding to C + G, T, G + A, respectively. These RSD values suggest a good homogeneity in the peak intensities, thus making QOS a reliable SERS sensor for gene expression detection.

The expression of OCT-4 based on SERS intensity ratio is displayed in Fig. 8d. The results suggest that the expression level of OCT-4 is upregulated in cancer cells when compared with fibroblast cells. Among the cancer cells, it can be observed that the self-renewal and stemness gene (OCT-4) is highly upregulated in lung cancer cells relative to breast cancer and pancreatic cells. Studies have proven the role of OCT-4 upregulation in self-renewal and initiating tumor by activating the downstream genes[42], leading to inhibition of cellular differentiation. The inhibition of cellular differentiation is positively correlated with the presence of cancer stem cells[45]. Hence, it can be concluded that DNA isolated from lung cancer cell exhibit an increased percentage of cancer stem cell marker compared to other cancer cells investigated. This observations are consistent with those reported in literature[43].

The expression of SOX-2 based on SERS intensity ratio is displayed in Fig. 8f. SOX-2 is consistently overexpressed in all cancer cells relative to fibroblast cells. The overexpression of SOX-2 is leads to increased cellular proliferation by cyclin D3 induction[46]. The up-regulation of SOX2 is associated with increased level of cancer stem cell markers. Altogether, the aberrant expression of SOX2 in pancreatic cancer and lung cancer impacts cellular proliferation, stemness and epithelial-

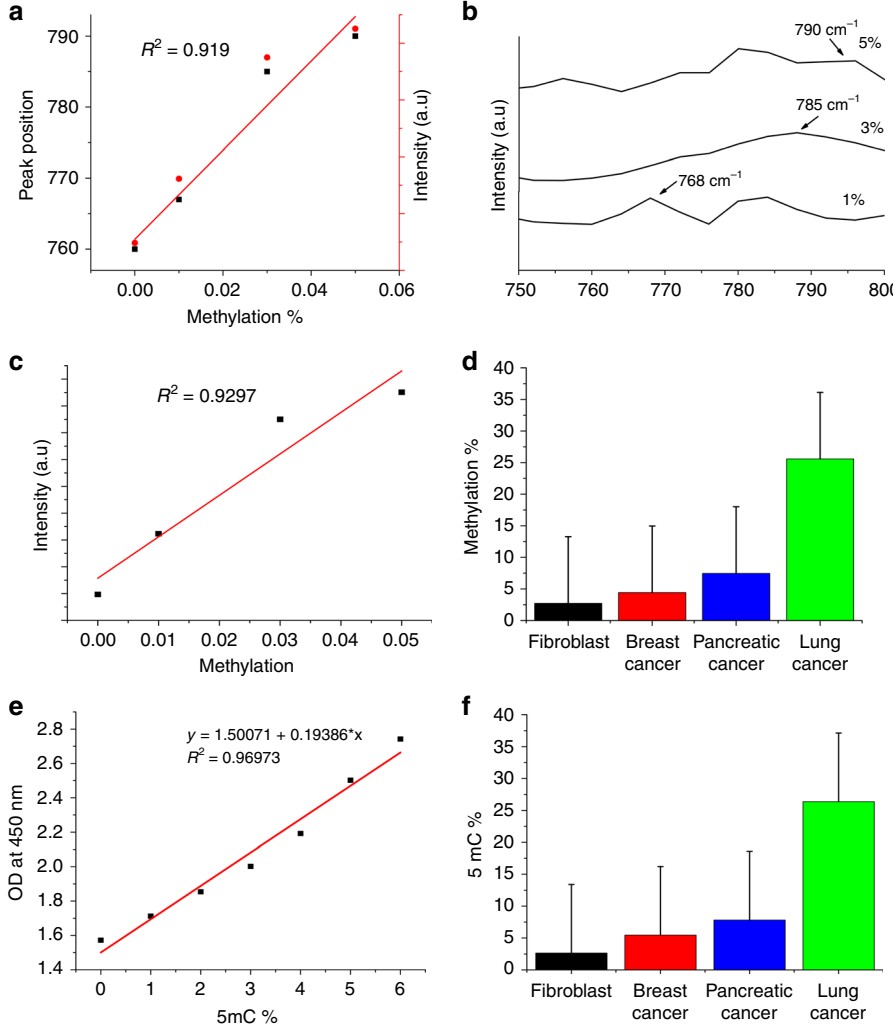

**Fig. 7 Quantification of global DNA hypermethylation using SERS. a** SERS spectra of 5mC DNA with different methylation percentages. **b** Standard curve based on peak position and SERS intensity for quantification of DNA methylation. **c** Standard curve based on peak intensity for quantification of DNA methylation. **d** Percentage DNA methylation obtained through QOS-based SERS. Validation of SERS based DNA methylation with colorimetric assay. **e** Standard curve based on OD at 450 nm. **f** Percentage DNA methylation based on colorimetric assay.

mesenchymal transition of cells thus regulating the formation of cancer stem cells[46,47]. The clinical implication of SOX2 upregulation includes poor prognosis, enhanced chemoresistance and decreased patient survival[43].

Studies have shown that aberrant expression of the genes namely OCT-4, SOX-2, c-Myc, Klf4 is essential to induce cellular reprogramming and cellular dedifferentiation[16]. Investigating the expression of these transcription factors forms only a small part of epigenetic landscape. Further, aberrant epigenetic alterations have proven to directly impact the clinical outcome of cancer therapy. Current methods of CSC identification and isolation relying on surface markers have proven to be inaccurate due to the presence of multiple markers on the primary tumor[48]. Therefore, analyzing the epigenetic alterations leading to changes in signaling pathways is expected to provide an accurate detection of CSCs.

The Fig. 9a–c shows the expression levels of GAPDH OCT-4 and SOX-2 in box plots. The data is represented using the box plot as the measurements are not normally distributed. It can be observed from Fig. 9a, that the expression levels of GAPDH shows a similar trend in all the 4 DNA samples. Fibroblast DNA shows lower expression of both OCT-4 and SOX-2 and lung

cancer DNA shows the highest expression level. PCR was performed, and the results agreed well with results from SERS detection. The PCR results in Fig. 9d confirms the SERS results. In addition, the PCR results obtained are from the amplified DNA, whereas the SERS results are from the unamplified PCR product. Hence, gene expression analysis using tag-free QOS sensor provides sensitive detection on par with PCR with a higher sensitivity and reliability which is demonstrated in Fig. 9e, f for relative expression levels of OCT-4 and SOX-2, respectively.

## Discussion

One of the key characteristics leading to abnormal and aggressive growth in cancer is genomic instability. Current methods to analyze the stability of a genome focusses on genome sequencing. Sequencing of the entire genome proves to be both time consuming and expensive. Additionally, sequencing for a single marker requires a template DNA specific for the genome in question. The requirement of a template DNA defeats the purpose of attaining universal genomic instability detection. Therefore, at least 4 tests are needed to get the complete base composition profile.

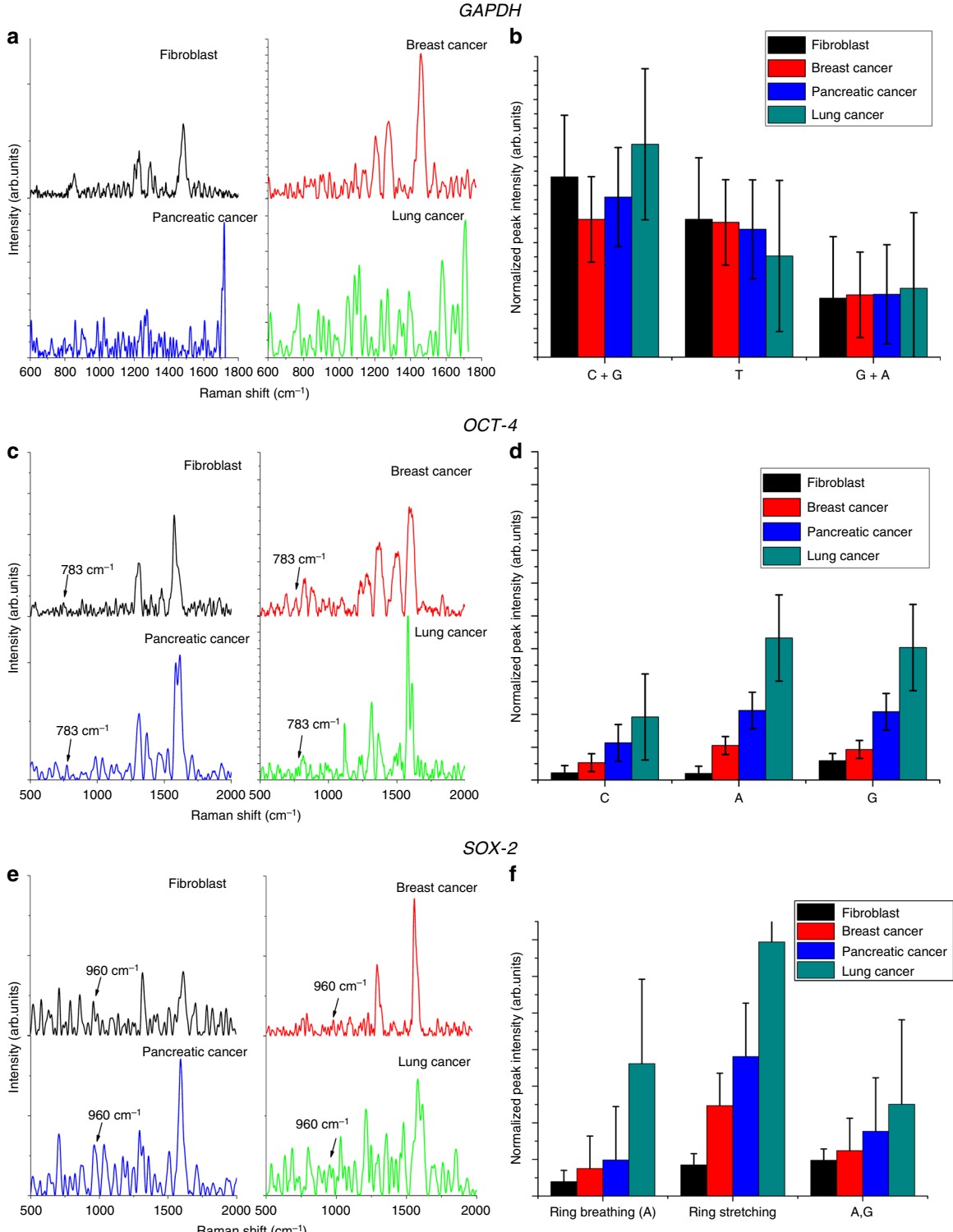

**Fig. 8 Gene expression detection using QOS. a**, **c**, **e** SERS spectra of GAPDH, OCT-4, and SOX-2, respectively, **b**, **d**, **f** expression ratio of GAPDH, OCT-4, and SOX-2, respectively.

The spectral analysis presented in this study are done with a single DNA molecule, corresponding to a concentration of fem-togram/μl of DNA. The ability to study single DNA molecule represents a highly sensitive and accurate detection method. Further, studies show that cancer is evolved from ultra-low concentrations of circulating tumor cells and cell free circulating tumor DNA[17]. Owing to the low concentration of these viable

tumor biomarkers, detecting single molecule is critical for the early detection of cancer.

The SERS spectra in Fig. 6, highlights the peak positions corresponding to the bases Adenine (A), Thymine (T), Cytosine (C + T), Phosphate backbone (PO2). A complete base composition is revealed in this single spectrum. in addition, DNA methylation markers are also present in the same spectra. It can

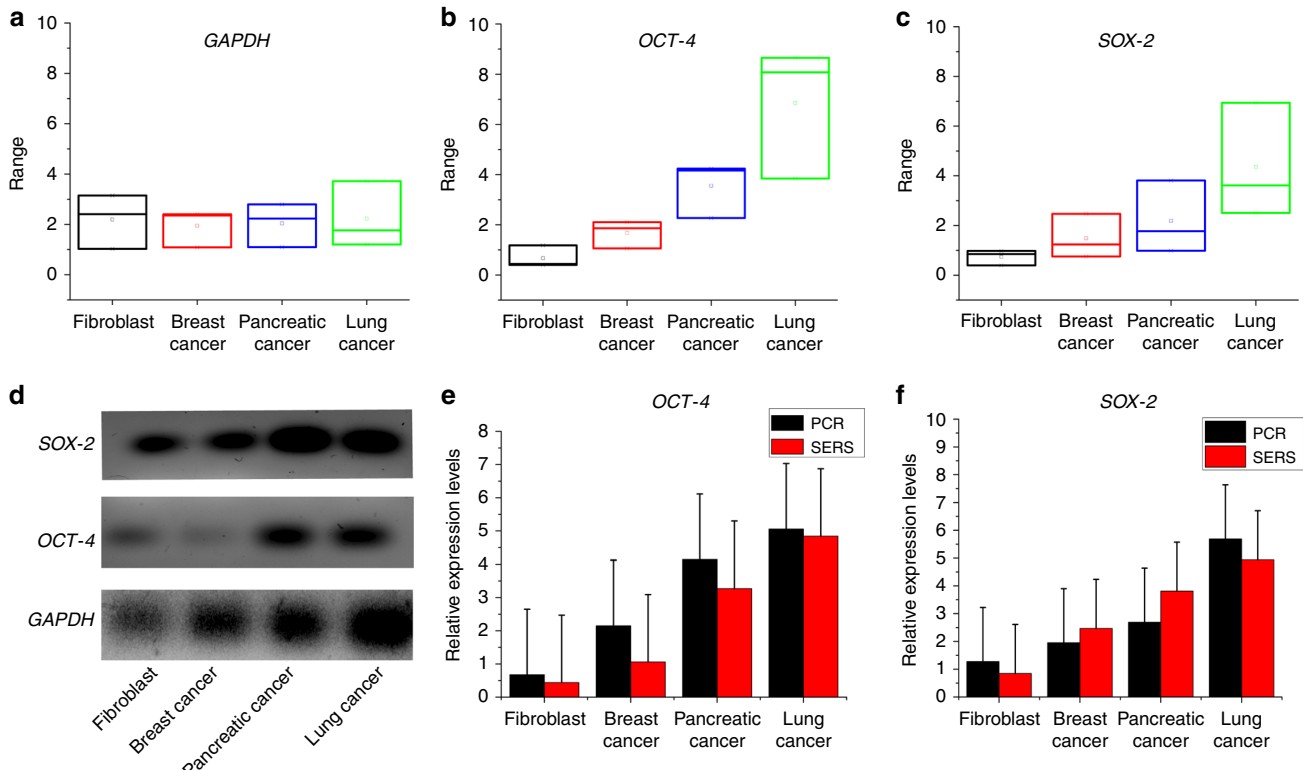

**Fig. 9 Validation of gene expression analysis by semiquantitative PCR. a–c** expression levels of GAPDH OCT-4 and SOX-2 in box plots, respectively. **d** Semiquantitative PCR to validate relative expression levels of OCT-4 and SOX-2 in comparison with GAPDH, **e**, **f** comparative data between relative gene expressions obtained from PCR and SERS using tag-free QOS.

be inferred that both base composition analysis and methylation analysis can be performed in a single test using tag-free QOS sensor.

Traditional techniques to study DNA methylation includes HPLC, mass spectrometry, pyrosequencing, bisulfite sequencing to name a few[11]. These methods suffer from disadvantages including the requirement of high purity of sample, high concentration of DNA in the range of microgram[13]. These bottlenecks make the traditional methods unfit for investigations in the single cellular level. Additionally, the majority of the traditional methods involve chemical modification of DNA[49], to obtain information about methylated regions of the genome, leading to alteration of native form of genomic DNA. The use of traditional methods is further impeded by the requirement of specialized software to interpret the results obtained.

Most recently techniques, such as electro-optical sensing[50], detection using nanopore technology[51], ELISA based assay[52] were developed to investigate DNA methylation. Although the recent techniques offer a high sensitivity compared to traditional ones, these methods require a template DNA/ label/ tag to analyze the methylation status, thus limits the applicability of these techniques to multiplex detection or to detect global methylation status. Besides, the ELISA based techniques can offer only an approximate evaluation of methylation status and are highly sensitive to contamination, resulting in a higher probability of false positive results[11].

Apart from studying methylation and structural variations in genomic DNA, investigating the stemness gene expression is essential to identify cancer stem cell phenotype. Traditional methods of gene expression involve polymerase chain reaction (PCR) to amplify and detect specific expression patterns. Although PCR remains to be the well accepted technique, it suffers from distinct disadvantages, such as modification of native

state of genomic DNA through amplification by strand extension, requires a high concentration, highly pure sample for efficient and accurate detection. Moreover, PCR techniques is only semiquantitative and does not provide a direct information about gene expression[15].

The designed QOS-based tag-free genomic DNA sensor can analyze genomic instability instantaneously and provide complete analysis in a single data acquisition. In other words, base composition and methylation are revealed in one test. Additionally, the tag-free genomic DNA sensor can be used to study multiple DNA methylation markers and has the capability to study global methylation status concurrently. The QOS sensor also offers clear advantages, including single-molecule sensitivity and detecting direct gene expression without amplification. In addition, QOS-based tag-free genomic DNA sensor does not alter the native structure of genomic DNA thus eliminating the need to disintegrate/fragment the genomic DNA to obtain useful information. Hence, the QOS-based tag-free genomic DNA sensor provides a tool for identifying the markers of cancer stem cells.

The perception on progression of cancer has changed due to advancement in single molecular genomic DNA analysis. Existing methods suffers from severe bottlenecks, including the use of fragmentation of DNA hence the loss of critical information, chemically altering the native state of DNA therefore inaccuracy of results. In addition, they are time-consuming and provide poor sensitivity. The use of genomic DNA detection using SERS provides significant new prospect for a new generation of cancer diagnostics. The QOS proposed in this research may lead to a rapid real-time diagnosis tool.

The QOS was synthesized by physical method using an ultrashort pulsed laser, thus shrinking organic semiconductor to quantum scale. The probe facilitates intense Raman enhancement, thereby ultra-sensitive molecular detection. In the QOS, the

mechanism of Raman enhancement is an interplay of surface plasmon of QOS probe activated by the presence high charge carrier density, charge transfer resonances and efficient exciton generation. The phenomenon of ultra-sensitivity of QOS was demonstrated using two Raman active molecules, Crystal Violet (CV) and Rhodamine 6G (R6G). The lowest reported limit of detection of $1 \times 10^{-15}$ M was achieved, substantiates the capability of the probe to be used for detection of ultra-low concentrations of analyte such as cancer biomarkers.

The ability of QOS for epigenetic analysis was demonstrated using genomic DNA isolated from four different cell lines, fibroblast cells (NIH3T3), breast cancer (MDA-MB 231), pancreatic cancer (AsPc-1), and lung cancer (H69-AR). Genomic DNA information including structural, molecular and gene expression aberrations were simultaneously obtained with one single tag-free detection of whole-strand native DNA. The molecular differences between the genomic DNA of cancerous and non-cancerous cells can then be analyzed by performing multivariate statistical analysis. The potential of QOS for gene expression analysis were validated with the tracing of two genes that are markers of cancer stem cells, namely Oct-4 and SOX-2. By using PCR assisted hybridization, we successfully demonstrated ultra-sensitive tag-free gene expression detection from the unamplified gene product. The shrinking of organic semiconductor pushes SERS sensitivity to single molecule, thus, opening the possibility to sense single-molecule genomic DNA without modification.

## Methods

**Synthesis of nanostructured QOS sensor**. The nanostructured network of quantum organic semiconductors was synthesized using a Clark-MXR IMPULSE pulsed Yb—doped fiber -amplified femtosecond laser to ionize a graphite substrate. The laser parameters were kept constant at a wavelength of 1030 nm with a circular polarization. The average laser power was maintained at 17 W and the laser pulse width of 214 fs was used for synthesizing nanostructured QOS. The graphite substrate was fixed on a piezo-driven system with a pattern of lines designed using EzCAD software. The pulse repetition rate of 4 MHz was used for generating the small QOS and the pulse repetition rate of 2 MHz was used to generate Large QOS. To introduce nitrogen atoms to the graphene lattice, gaseous nitrogen at low pressure was introduced into the laser ion plume. The injected nitrogen gas was evenly surrounded using individual nozzles in the ionization zone at a constant flow rate.

**Characterization of nanostructured QOS sensor**. The structure of the QOS network was imaged using a FEI Quanta FEG 250 scanning electron microscope. The morphology of individual QOS was analyzed using a Hitachi H-7000 HRTEM on copper mesh grids. The images obtained were used to obtain the size distribution of QOS. The size and size distribution were measured using the ImageJ software. The XPS measurement was obtained using a Thermo fisher K alpha XPS system using an Al Kα X-ray source. The quantification was performed using Avantage software.

The optical characterization Raman spectra was obtained using Renishaw Invia Confocal Raman spectrometer equipped with a Leica DMI6000 epifluorescence microscope. The wavelength of 785, 532 nm was used to obtain Raman spectra for characterizing QOS. The UV-vis spectrum was obtained using Shimadzu UV-3600 UV-Vis-NIR spectrophotometer. To obtain the spectrum, the QOS was ultrasonicated with water. The UV-visible transmission spectra were used to analyses the interaction between QOS and analyte molecule.

**SERS spectral analysis of Raman probe molecules**. To acquire the Raman spectra of Crystal violet on the QOS and graphite substrate a Renishaw Invia Confocal Microscope with a ×20 magnifying lens was used. The Raman spectra was obtained at 785 nm. The acquisition time was kept constant at 10 s and three acquisitions was made for each spectrum to ensure reproducibility and uniformity of signals. In all, 10 µl of the specified concentration of crystal violet solution was applied to QOS prior to collection of Raman spectra. For each sample, at least 20 spectra were obtained from different locations of substrate to collect spectra with a good signal-to- noise ratio and reproducibility.

The enhancement factor was calculated according to the calculations provided in the literature[1,53–55]. The enhancement factor per molecule is calculated using the

equation:

$$EF = \frac{\frac{I_{QOS}}{N_{QOS}}}{\frac{I_{graphite}}{N_{graphite}}}$$

$$N_{graphite} = \pi r^2 hc N_A$$

$$N_{QOS} = SA_{eff}\left(C_{ads}N_A \frac{1000\,L}{1\,m^3}\right)^{2/3}$$

where I is the intensity of the peak selected, N is the number of molecules in the laser spot. The number of molecules in the laser spot size was calculated using a laser spot size of 1 µm (0.5 µm radius), the laser penetration depth was considered as 10 µm based on ×20 long working distance objective. Hence, the volume in the laser spot size 26.17 µm$^3$. Then by using the values for crystal violet (density 0.981 g/cm$^3$, molar mass 407.979 g/mol) and Rhodamine 6 G (density 1.26 g/cm$^3$, molar mass 479.02 g/mol) to find the number of molecules in the laser spot size. Using the above-mentioned values, the values of $N_{graphite}$ as $1.047 \times 10^{16}$, $1.579 \times 10^{16}$ for crystal violet and rhodamine 6 G, respectively. The number of molecules contributing to SERS enhancement on the surface of QOS is based on the following assumptions: (i) The first layer of QOS contributes to observed SERS enhancement, (ii) The molecules adsorbed on the surface of QOS lie parallel to the surface. This is confirmed by the presence of symmetric modes in the SERS spectra including out-of-plane C–H deformation, ring deformation, in-plane C–H bending vibration, ring stretch vibration[54]. This assumptions is further confirmed by DFT calculations provided in the literature[56]. The value of $N_{QOS}$ was obtained by dividing the laser surface area (6.28 µm$^2$) by the effective cross-sectional area per molecule ($6 \times 10^{-7}$)[54,57,58]. Hence, the calculated value of $N_{QOS}$ to obtain the number of molecules on QOS $1.04 \times 10^{-7}$. The values of $I_{QOS}$ and $I_{graphite}$ was obtained from the SERS spectra in Fig. 2. The calculation for picomolar concentration of analyte is provided in Table 1.

**Genomic DNA isolation and SERS analysis**. NIH3T3 (Fibroblast), MDA-MB-231 (Breast cancer), AsPc-1(Pancreatic Cancer), H69AR (Lung Cancer) cells obtained from ATCC are cultured separately using DMEM containing 10% FBS, 1% pen/strep at 37 °C in 5% CO$_2$ atmosphere. The cells were split in the ratio of 1:3 on reaching confluence. The cells were harvested using Trypsin-EDTA solution, centrifuged at 1000 rpm for 5 min to pellet the cells. The genomic DNA was isolated according to the manufacturer's protocol. The concentration and purity of the isolated genomic DNA was measured using nanodrop. To obtain the SERS spectra, 10 µl of genomic DNA was placed on QOS and irradiated with 785 nm Raman laser. To ensure the reproducibility and uniformity of signals the acquisition time was kept constant at 10 s and 3 acquisitions were made. The SERS peaks used for analysis the peak intensities used for biological analysis was carefully chosen with the signal- to-noise ratio higher than three times the standard deviation of the data. Gene expression detection was performed using PCR assisted hybridization. The genomic DNA was only hybridized, and the unamplified sample was used for SERS detection of OCT-4 and SOX-2 expression.

**Statistical analysis**. All the statistical analysis were performed using XLSTAT software[59]. We used two-tailed student's t test to compare the differences between the DNA bases, global hypermethylation levels, relative gene expression levels. We applied ANOVA (F-test) to compare the differences between the DNA isolated from various cellular models. The $P < 0.05$ was considered statistically significant in all the analysis reported in the paper.

**Reporting summary**. Further information on research design is available in the Nature Research Reporting Summary linked to this article.

### Table 1 Detailed calculation values of enhancement factor.

|  | CV | R6G |
|---|---|---|
| Spot volume (µm$^3$) | 26.17 | 26.17 |
| Density (g/µm$^3$) | 9.81E + 11 | 1.26E + 12 |
| molar mass (g/mol) | 407.979 | 479.02 |
| $N_{graphite}$ | 1.0474E + 16 | 1.57953E + 16 |
| SA of spot (µm$^3$) | 6.28 | 6.28 |
| effective cross section | 0.0000006 | 0.0000006 |
| $N_{QOS}$ | 10466666.67 | 10466666.67 |
| $N_{graphite}/N_{QOS}$ | 1.635E + 18 | 1509105218 |
| $I_{QOS}/I_{graphite}$ | 1106.048311 | 3026.804331 |
| EF | 1.80839E + 12 | 4.56777E + 12 |

## Data availability

The data that support the findings of this study are available from the corresponding author upon reasonable request.

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

## Acknowledgements

This research was funded by NSERC Discovery Grant 132950, 134361.

## Author contributions

S.G., K.V., and B.T. worked together in designing the project, S.G. performed the experiments and wrote the paper, K.V. and B.T. assisted in results, discussion, and edited the paper.

## Competing interests

The authors declare no competing interests.
