## [Peer Review File · Nature Communications]

Reviewers' comments:

Reviewer #1 (Remarks to the Author):

In this article the authors demonstrate two important advances in the practice of surface-enhanced Raman scattering applications. They take advantage of the special properties of organic semiconductors (as pioneered by Yilmaz et al. Refs. 15, 16) and add the ability to excite plasmon resonances in the substrate by reducing the particle size to the quantum confinement limit. This results in increasing electron density, moving the plasmon resonance to higher energies (i.e. closer to the laser excitation) thereby increasing the SERS enhancement. Thus, they are able to take advantage of the additional plasmon-charge-transfer resonance coupling. They claim enhancement factors of the order of 10^{12} , with molecules such as R6G and Crystal Violet. These are favorites among SERS researchers due to their inherently high enhancement factors. Furthermore, enhancements of this magnitude moves into the single-molecule detection regime. While this is possible, single molecule detection usually involves special detection techniques (due to spectral blinking, and rare hotspot formation). Therefore, in the interest of scientific caution, I recommend the authors display more modesty in their claims. Nonetheless, this remains a remarkable achievement.

The second important advance is the use of statistical techniques to extract meaningful spectra from DNA cancer cell samples. There is increasing utilization of PCA and other similar procedures (such as linear discriminant analysis (LDA)) to enhance our ability to detect and analyze large molecules, or mixtures of molecules which would ordinarily have such a dense Raman spectrum as to be un-analyzable by normal Raman techniques. The authors are to be commended to venture into this area.

Allow me to dispute one aspect of this work. That is the use of the term EMERS for their observations. Almost all the early SERS work involved metal nanoparticles, which provided a surface plasmon resonance important for the effect. However, the recent enormous expansion of SERS to semiconductor substrates has shown that even when the plasmon resonance is far from the laser, other resonances can be marshalled to provide a considerable enhancement (as well as other important advantages). The authors have shown a way to induce metal-like properties in semiconductors by reducing the particle size. It is confusing and possibly disingenuous to make up a new term for a rather old phenomenon.

Reviewer #2 (Remarks to the Author):

1. Key results: Please summarise what you consider to be the outstanding features of the work.

The authors claim that they have developed a SERS system based on organic semiconductors to distinguish between the molecular signature of cancer stem cells and genomic DNA. They demonstrate large SERS enhancements in the order of 10^{12} using this system. How does this compare to SERS enhancement obtained from plasmonic metals? Considering this, what is the benefit of using organic SERS for biological analysis. Aren't metallic probes more robust?

2. Validity: Does the manuscript have flaws which should prohibit its publication? If so, please provide details.

The biggest issue with the manuscript is that it does not provide sufficient amount of control experiments and validation bioassays to support the claims made here. For example to validate their ability to identify methylated bases, they need to compare their results to those obtained using conventional sequencing methods.

Additionally, gene expression levels are not validated using methods like microarrays or quantitative PCR.

The authors claim a sub-femtomolar limit of detection; however, this calculation is not done using the methods published in the literature considering signal noise, blank, and a linear slope. Also the calibration curve only includes three data points. This plot should include a data point every decade to support log-linearity.

The authors discuss efficient charge transfer between the analyte and QOS as one of the two mechanisms contributing to enhancement. To support this, they present a UV/VIS spectrum; however, they do not provide the UV/VIS of the CV or RGB alone to understand the reasoning behind the spectra obtained in QOS+RGB and QOS+CV. Not enough information is presented for the differential capacitance measurements to explain the charge transfer mechanism.

3. Originality and significance: If the conclusions are not original, please provide relevant references. On a more subjective note, do you feel that the results presented are of immediate interest to many people in your own discipline, and/or to people from several disciplines?

The results, if properly presented and validated, are significant.

4. Data & methodology: Please comment on the validity of the approach, quality of the data and quality of presentation. Please note that we expect our reviewers to review all data, including any extended data and supplementary information. Is the reporting of data and methodology sufficiently detailed and transparent to enable reproducing the results?

The figures are poorly presented and are very difficult to understand. The formatting is inconsistent across multiple figures and axes titles are missing or difficult to read.

5. Appropriate use of statistics and treatment of uncertainties: All error bars should be defined in the corresponding figure legends; please comment if that's not the case. Please include in your report a specific comment on the appropriateness of any statistical tests, and the accuracy of the description of any error bars and probability values.

Appropriate statistical analyses are not done to understand whether methylated and non-methylated bases are statistically significant between different bases. For example in Figure 8b.

6. Conclusions: Do you find that the conclusions and data interpretation are robust, valid and reliable?

As mentioned earlier, lack of quantitative validation data for data presented here by SERS significantly weakens the validity of the conclusions

Reviewer #3 (Remarks to the Author):

In this article the authors examine the use of an organic semiconductor dot having very small size (3-7 nm) as a probe/substrate for surface-enhanced Raman spectroscopy (SERS). They synthesize carbon-dots as a SERS probe/substrate using femtosecond laser pulse irradiate on a graphite substrate in the presence of nitrogen gas. They use R6G and CV as test molecules. The authors claim up a $\sim 10^{12}$ enhancement of Raman intensities. As a real life application, they utilize synthesized organic semiconductor dots in the detection of cancer stem cell marker OCT4 and SOX2. Although graphene and graphene-based quantum dots have been utilized for many years (especially for the last 3 years) as SERS substrate, this is the first use, to my knowledge, of a graphene-based quantum dot in the detection of CSC markers. It is here that this paper is interesting. Unfortunately, the authors seem quite unaware of the existing experiments (Nat. Commun. 2018, 9, 193; Nanoscale 2016, 8, 8863 etc.). Therefore, the SERS substrates they fabricated are not novel. The authors propose a new SERS enhancement mechanism called electron mobility enhanced Raman. However, their discussion is mainly based on speculation (there is no experimental data to support their claims). They also claim that the record enhancement factor of 10^{12} is due to the electromagnetic enhancement which is originated from the surface plasmons of organic semiconductor dots. Again, there is no experimental proof to support this point. Therefore, I am skeptical about the interpretations and the proposed enhancement mechanism. The conclusions are mostly suggestive and it is difficult to follow. Besides these important points, there are also some technical issues that must be addressed. For the EF calculations, the authors use picomolar (10^{-12} M) concentrations of

analyte molecules. I think that's why they observed such a high EF. If they used conventional concentrations such as 10^{-3} or 10^{-5} M, they would get much lower EF.

In sum, I consider this an interesting work. However, I see serious issues which affect both parts (fabrication and application) thus preventing publication if not properly addressed. I suggest to the authors to revise their work before submitting to another journal.

Minor points:

- In fig4, there is no Raman spectra of analytes on bare graphite. Moreover, there are some shifting in peak positions unlike authors claim.
- I do not understand the sentence in page 15 "The electronegative oxygen functional groups in the 3D architecture creates a local electric field on the analyte molecules, thus enabling a hotspot formation upon laser excitation". How oxygen atoms create electromagnetic hot spots?
- In Fig 6, 7, 8 and 9, the signal to noise ratio seems very low and it is hard to use these Raman spectra in the reliable detection of biological molecules.
- There is also no data about the reproducibility of the fabricated substrates/probes in SERS which is important for real life applications.

Response to reviewers

With reference to your letter regarding revision of the manuscript titled “Shrinking Organic Semiconductors to Quantum Scale – Utilizing Genomic DNA for Cancer Stem Cell Detection - NCOMMS-19-10288” the authors would like to thank the reviewers for the encouraging comments on the novelty of the work. To further provide a better clarity on the concept and to address the reviewer’s comments the following new experimental data has been added. In addition, to the new experimental data extensive analysis was performed to further improve the understandability of the manuscript. Some of the major revisions are as follows:

The following changes has been made in the manuscript:

1. Based on the reviewer’s comments, we have incorporated new experimental data on SERS spectrum of CV and R6G on bare graphite substrates as a control experiment in figure 4 (Reviewer #3)
2. New experimental data on the linear relationship between peak intensity and analyte concentration was added to support the log-linearity. In addition, experimental data on micromolar concentration of analyte was added. The data is incorporated in figure 4. (Reviewer #2)
3. New experimental data on SERS spectra of micromolar and nanomolar concentration of CV and R6G was performed and the data is incorporated in figure S1 in the supplementary information of the manuscript (Reviewer #3, Reviewer #2)
4. New experimental data on reproducibility of SERS signals was acquired for tag free QOS sensor at single molecule level (10^{-15} M). A new section on reproducibility of SERS signals at was integrated in the manuscript in figure 5. (Reviewer #3, Reviewer #2)
5. New experimental data on UV-visible transmission measurements of CV and R6G was incorporated in figure 6 to provide a better understanding on the mechanism of charge transfer in the QOS-analyte system. (Reviewer #2)
6. To provide a better comprehension on the role of enhanced electron mobility on SERS enhancement in quantum organic semiconductors, elaborate discussion on the differential conductance data is added to the manuscript. (Reviewer #3, Reviewer #2)
7. New experimental data was acquired for the SERS spectra of genomic DNA presented in figure 7. The spectra were collected at 20 different points and the average spectra is

presented in figure 7 resulting in a higher signal to noise ratio. Similar approach was followed for the SERS spectra provided for gene expression analysis in figure 11. (Reviewer #3, Reviewer #2)

8. New experimental data on standardization and quantification of global DNA hypermethylation was reinforced in figure 10. (Reviewer #2)
9. New biological assays were carried out to validate the sensitivity and reliability of QOS as a tag free genomic DNA sensor for detection of global DNA hypermethylation. The data is incorporated in figure 10. Further, the comparison of DNA methylation values obtained from both SERS and validation assay is presented in table 2. (Reviewer #2)
10. New experimental data was added for gene expression detection using tag-free QOS sensor. New data on SERS sensing of Gene expression analysis of GAPDH acting as an internal standard is incorporated in figure 11. (Reviewer #2)
11. In addition, the gene expression analysis was validated with semi-quantitative PCR, the data is incorporated in figure 12. (Reviewer #2)
12. The figures in the manuscript have been revised and the axes have been made clear to provide better clarity and understanding of the manuscript. (Reviewer #2)

Point by point response to reviewers

Response to Reviewer 1:

Comment 1: In this article the authors demonstrate two important advances in the practice of surface-enhanced Raman scattering applications. They take advantage of the special properties of organic semiconductors (as pioneered by Yilmaz et al. Refs. 15, 16) and add the ability to excite plasmon resonances in the substrate by reducing the particle size to the quantum confinement limit. This results in increasing electron density, moving the plasmon resonance to higher energies (i.e. closer to the laser excitation) thereby increasing the SERS enhancement. Thus, they are able to take advantage of the additional plasmon-charge-transfer resonance coupling. They claim enhancement factors of the order of 10^{12} , with molecules such as R6G and Crystal Violet. These are favorites among SERS researchers due to their inherently high enhancement factors. Furthermore, enhancements of this magnitude move into the single-molecule detection regime. While this is possible, single molecule detection usually involves special detection techniques (due to spectral blinking, and rare hotspot formation). Therefore, in the interest of scientific caution, I recommend the authors display more modesty in their claims. Nonetheless, this remains a remarkable achievement.

The second important advance is the use of statistical techniques to extract meaningful spectra from DNA cancer cell samples. There is increasing utilization of PCA and other similar procedures (such as linear discriminant analysis (LDA)) to enhance our ability to detect and analyze large molecules, or mixtures of molecules which would ordinarily have such a dense Raman spectrum as to be un-analyzable by normal Raman techniques. The authors are to be commended to venture into this area.

Response: The authors thank the reviewer for such encouraging and positive comments. Thanks for highlighting the novelty of work.

Comment 2: Allow me to dispute one aspect of this work. That is the use of the term EMERS for their observations. Almost all the early SERS work involved metal nanoparticles, which provided a surface plasmon resonance important for the effect. However, the recent enormous expansion of SERS to semiconductor substrates has shown that even when the plasmon

resonance is far from the laser, other resonances can be marshalled to provide a considerable enhancement (as well as other important advantages). The authors have shown a way to induce metal-like properties in semiconductors by reducing the particle size. It is confusing and possibly disingenuous to make up a new term for a rather old phenomenon.

Response: The authors thank the reviewer for such encouraging and positive comments. Based on the reviewer's comment, we have made the following changes in the manuscript to provide a better clarity.

1. The authors have removed the new term EMERS and rephrased in the abstract, introduction and conclusion.
2. The authors have used the inherent property of high charge carrier mobility of organic semiconductor to explain the efficient charge transfer mechanism in the QOS-analyte system leading to a high SERS enhancement.

To further provide clarity in the manuscript, the following changes has been incorporated in the manuscript:

The following section was rephrased in the abstract in page 2 [line 17 – 26]

“Despite the many advantages it has to offer, their application is limited due to its inability for ultra-sensitive detection. The major obstacle is that organic semiconductor Raman excitation is dominated by, a weak SERS mechanism, charge transfer, making high enhancement efficiency hard to achieve. By approaching quantum scale, the phenomenon of surface plasmon is activated. Exciton resonance in organic semiconductor leads to increase in Raman cross section. It offers strong coupling of electronic and photonic modes allowing efficient charge transfer enabling an effective single molecule detection with an incomparable enhancement factor of 10^{12} on a bare quantum organic semiconductor which is a 9-order increase in enhancement efficiency compared to existing organic probes.

The following section was rephrased in the introduction in page 5 [line 22 – 41] and page 6 [line 1-5]

“In this article we report a tag free QOS sensor aimed to detect cancer stem cells by utilizing single molecule genomic DNA. The genomic DNA holds the key information about the

structural, molecular and genetic modifications which can be an effective marker for cancer stem cells. The ultrashort pulsed laser processing in the presence of nitrogen gas enabled the shrinking of organic semiconductor to quantum scale which resulted in increased charge carrier mobility essential for efficient charge transfer necessary for SERS enhancement. Detection of CV and R6G demonstrated a label-free single-molecule detection (femtomole) sensitivity with an unmatched enhancement factor of 10^{12} , which is 9 orders higher than reported value from organic semiconductors. These results motivated us to test QOS for the detection of epigenetic markers of cancer stem cells. Studying the epigenetic origin of cancer stem cells sets a new paradigm towards the fundamental understanding of cancer immortality. The genomic DNA isolated from various cellular models were used to determine the structural and molecular changes tag free QOS sensor. The practical applicability of tag free QOS sensor was validated using genomic DNA isolated from 4 different cellular models, namely fibroblast cells (NIH3T3), Breast Cancer (MDA-MB 231), Pancreatic Cancer (AsPc-1) and Lung Cancer (H69-AR). Base composition of DNA and methylation markers are collected with one single detection. By performing multivariate statistical analysis, we could determine the molecular differences between the genomic DNA of cancerous and non-cancerous cells. Next, QOS was studied for gene expression analysis. The expression of 2 genes critical for regulation of cellular senescence was implemented to attain a comprehensive understanding of the epigenetic landscape. The ability of QOS to study the epigenetic origin of cancer stem cells, has opened the new possibility of extending the utilization for non-invasive, personalized cancer diagnosis. This study opens up new possibilities with organic sensors. It holds potential not only for disease biomarkers detections but also other applications requiring ultra-sensitive detection, such as rapid sensing of environmental contaminants, hazardous chemicals and explosives.”

The following section was rephrased in the conclusion in page 34 [line 29- 37]

“The QOS was synthesized by physical method using an ultrashort pulsed laser, thus shrinking organic semiconductor to quantum scale. The probe facilitates intense Raman enhancement, thereby ultra-sensitive molecular detection. In the QOS, the mechanism of Raman enhancement is an interplay of surface plasmon of QOS probe activated by the presence high charge carrier density, charge transfer resonances and efficient exciton generation. The phenomenon of ultra-sensitivity of QOS was demonstrated using two Raman active molecules, Crystal Violet (CV)

and Rhodamine 6G (R6G). The lowest reported limit of detection of 1×10^{-15} M was achieved, substantiates the capability of the probe to be used for detection of ultra-low concentrations of analyte such as cancer biomarkers.”

Response to Reviewer 2:

The authors thank the reviewers for the encouraging comments on the novelty of the work. To further provide a better clarity on the concept and to address the reviewer's comments the following new experimental data has been added.

1. New experimental data on reproducibility of SERS signals was acquired for tag free QOS sensor at single molecule level (10^{-15} M). A new section on reproducibility of SERS signals at was integrated in the manuscript in figure 5.
2. New experimental data on UV-visible transmission measurements of CV and R6G was incorporated in figure 6 to provide a better understanding on the mechanism of charge transfer in the QOS-analyte system.
3. New biological assay to validate the global hypermethylation detection using tag free QOS was performed. The data analysis is reinforced in figure 10 e, f (page 28)
4. Based on the validation data, new data gathering, and analysis was performed to standardize and quantify DNA global hypermethylation in genomic DNA samples of fibroblast, lung cancer, pancreatic cancer, breast cancer.
5. New experimental data on SERS spectra of methylated DNA (1%, 3%, 5% methylation) was performed and is incorporated in figure 10b to provide the standard curve for global hypermethylation analysis which aided in quantification of methylation status in the genomic DNA samples.
6. Additionally, to point out the validity and sensitivity of tag free QOS sensor to detect global DNA hypermethylation status of genomic DNA, the data obtained from the validation assay and SERS assay is presented in the results section as table 1 (page 29).
7. New experiment on PCR was performed to validate gene expression detection using tag free QOS.
8. The authors have also incorporated new experimental data by introducing an internal standard for gene expression detection. The gene GAPDH was used as the internal standard.
9. New experimental data on the SERS spectra of GAPDH is incorporated in figure 11.

10. Additional data analysis on determination of relative gene expression levels in genomic DNA samples of fibroblast, lung cancer, pancreatic cancer, breast cancer, was performed and incorporated in figure 12.
11. In addition, to highlight the validity and sensitivity of tag free QOS sensor to detect gene expression using unamplified gene product, the similarity between the relative gene expression levels obtained from SERS and PCR is reinforced in figure 12 e, f for OCT-4, SOX-2 respectively.

The point – to – point response is as follows:

Comment 1: The authors claim that they have developed a SERS system based on organic semiconductors to distinguish between the molecular signature of cancer stem cells and genomic DNA. They demonstrate large SERS enhancements in the order of 10^{12} using this system.

Response: The authors thank the reviewer for the encouraging comments.

Comment 1.1: How does this compare to SERS enhancement obtained from plasmonic metals?

The authors thank the reviewer for the valuable insight. The SERS enhancement achieved using the tag free QOS (Quantum Organic Semiconductor) sensor is on the same order of that exhibited by plasmonic materials. The following section has been added to the results section manuscript page 17 (line 6 – 10) to provide more clarity on the SERS enhancement factor obtained from tag free QOS:

“In the case of QOS used in this study, the achieved high enhancement factor for bare organic semiconductor is in the order of 10^{12} . The enhancement factor obtained using QOS show similar values as exhibited by conventional SERS substrates like gold, silver. Furthermore, both the analyte molecules (CV, R6G) exhibited similar enhancement factor values indicating the dependence of SERS enhancement mechanism is dependent on multiple enhancement mechanism”

Comment 1.2: Considering this, what is the benefit of using organic SERS for biological analysis.

The authors thank the reviewer for the valuable insight. Compared to their inorganic counterparts, organic probes exhibit a higher biocompatibility. Furthermore, organic

semiconductor materials are stable in cell culture medium which makes them ideal for cellular based diagnostic applications. In addition, carbon-based organic semiconductor materials have proven to be chemically stable thus remaining unreactive inside the cellular environment.

The following paragraph was added to the introduction page 5 (line 5-10) to further emphasize on the benefits on using organic SERS for biological analysis:

“Due to these limitations, the use of organic semiconductors as SERS sensor is scarce, although organic semiconductors offer many attractive advantages than its inorganic counterparts in terms of biocompatibility. In addition, carbon-based organic semiconductor probes are chemically stable in the cell culture medium^[1] and remain inactive in the cellular microenvironment^[2]. These properties make organic semiconductors uniquely qualified to be used as a biological sensor.”

Comment 1.3: Aren't metallic probes more robust?

The authors thank the reviewer for the valuable insight. Despite showing a high enhancement efficiency, the metallic probes suffer from severe limitations including inconsistent spectral signature owing to the predominance of hot spots, uneven particle aggregation in a colloidal state and limited formation of hotspots in the detection zone resulting in unreliable for detecting single molecules. Hence, there is always a constant search for new class of materials to replace metallic probes. Organic semiconductors have demonstrated the potential to replace metallic probes as a reliable SERS sensor.

The following paragraph was added in the introduction in page 4 (line 32 – 38) to highlight the importance of organic semiconductor probes:

“However, current research on SERS have been largely dominated by plasmonic materials. Despite providing a high enhancement efficiency, the conventional probes suffer from inherent disadvantages such as poor stability, poor biocompatibility, complex synthesis process along with high cost of fabrication^[3]. Hence, there is always a constant search for new class of materials to replace metallic probes. Organic semiconductors have demonstrated the potential to replace metallic probes as a reliable SERS sensor.”

The following separate section was added to highlight the reproducibility and reliability of SERS detection by QOS (Quantum Organic Semiconductor) in page 17

“Repeatability and Reproducibility of SERS enhancement:

Figure 5: Reproducibility analysis of SERS signals of a, b) CV , R6G on QOS respectively on QOS, c) intensities of characteristic CV peak at 1613 cm⁻¹ shown in spectra (a) error bars: SD, d) intensities of characteristic R6G peak at 1643 cm⁻¹ shown in spectra (b) error bars: SD.

In the traditional metal-based SERS system, repeatability and reproducibility of SERS signals are the major bottlenecks. Hence, we tested the reproducibility of the obtained SERS signals by measuring the SERS spectra at 5 different points. The obtained spectra are shown in figure 5 a, b for CV and R6G respectively. It can be observed from figure 5a showing consistent reproducible

characteristic SERS peak of CV at 1613 cm^{-1} . The corresponding peak intensity shown in figure 5c at 5 different points show a little difference with a relative standard deviation of 1.68%, thus confirming excellent reproducibility. The similar results for R6G is shown in figure 5b. The spectra showcase a consistent peak at 1643 cm^{-1} and corresponding uniform peak intensity with an RSD of 2.92% is shown in figure 5d. The presence of uniform intensity with a low RSD value indicates the reliability of nanostructured QOS as a SERS substrate.”

Comment 2: The biggest issue with the manuscript is that it does not provide sufficient amount of control experiments and validation bioassays to support the claims made here. For example, to validate their ability to identify methylated bases, they need to compare their results to those obtained using conventional sequencing methods. Additionally, gene expression levels are not validated using methods like microarrays or quantitative PCR.

Response: The authors thank the reviewer for the encouraging comments. The authors have added the following validation bioassays based on the reviewer’s comments.

1. New biological assay to validate the global hypermethylation detection using tag free QOS was performed. The data analysis is reinforced in figure 10 e, f (page 28)
2. Based on the validation data, new data gathering, and analysis was performed to standardize and quantify DNA global hypermethylation in genomic DNA samples of fibroblast, lung cancer, pancreatic cancer, breast cancer.
3. New experimental data on SERS spectra of methylated DNA (1%, 3%, 5% methylation) was performed and is incorporated in figure 10b to provide the standard curve for global hypermethylation analysis which aided in quantification of methylation status in the genomic DNA samples.
4. Additionally, to point out the validity and sensitivity of tag free QOS sensor to detect global DNA hypermethylation status of genomic DNA, the data obtained from the validation assay and SERS assay is presented in the results section as table 1 (page 29).
5. New experiment on PCR was performed to validate gene expression detection using tag free QOS.
6. The authors have also incorporated new experimental data by introducing an internal standard for gene expression detection. The gene GAPDH was used as the internal standard.

7. New experimental data on the SERS spectra of GAPDH is incorporated in figure 11.
8. Additional data analysis on determination of relative gene expression levels in genomic DNA samples of fibroblast, lung cancer, pancreatic cancer, breast cancer, was performed and incorporated in figure 12.
9. In addition, to highlight the validity and sensitivity of tag free QOS sensor to detect gene expression using unamplified gene product, the similarity between the relative gene expression levels obtained from SERS and PCR is reinforced in figure 12 e, f for OCT-4, SOX-2 respectively.

Comment 2.1: Validate their ability to identify methylated bases, they need to compare their results to those obtained using conventional sequencing methods.

The authors thank the reviewer for providing this constructive comment. The authors have incorporated the corresponding validation assay to support the ability of tag free QOS sensor to detect global hypermethylation in Genomic DNA. The authors have also reinforced a separate section on standardization and validation of DNA methylation status in the results section. Further, we have also incorporated a standard curve using 1%, 3%, 5% methylated standard to provide further insight on the quantification of global DNA hypermethylation. Validation of global DNA hypermethylation was performed using Global DNA methylation identification kit. The results obtained from SERS detection is comparable to the validation assay. To provide a clear understanding the following section is added to the manuscript in the results in page 27 - 29

“Standardization and Validation of Methylation analysis:

The ability of QOS to detect molecular changes in DNA further enabled us to investigate the ability to quantify global DNA methylation percentage. Figure 10 a showcases the SERS spectra of 5mC DNA with different methylation percentages. It can be inferred from figure 10 b that as the methylation percentage increases, there is a significant red shift in peak position accompanied with a significant increase in peak intensity. The standard curve shown in figure 10 c shows a linear correlation between peak intensity and methylation percentage. Hence, based on the linear fit obtained, the quantification of DNA methylation is presented in figure 10 d. Based on the linear fit of standard curve, the methylation percentage in sample DNA was calculated with the equation: $y = 1572 + 154661 * x$. The results (figure 10d) show that SERS based DNA methylation detection was as good as conventional detection systems. It should also be noted that

with the QOS based SERS sensor the detection time is greatly reduced with an improved sensitivity

Standardization of Methylation assay

Validation assay

Figure 10 : Quantification of global DNA hypermethylation using SERS a) SERS spectra of 5mC DNA with different methylation percentages b) Standard curve based on peak position and SERS intensity for quantification of DNA methylation c) Standard curve based on peak intensity for quantification of DNA methylation d) % DNA methylation obtained through QOS based SERS. Validation of SERS based DNA methylation with colorimetric assay e) standard curve based on OD at 450nm f) %DNA methylation based on colorimetric assay

To validate proposed method of SERS to detect and quantify methylation levels, we performed the colorimetric global DNA methylation kit obtained from ABCAM (AB233486). The assay was performed according to the manufacturer’s protocol and the results are presented in figure 10 e, f. The standard curve for the colorimetric assay is presented in figure 10e. The methylation percentage obtained from the colorimetric assay for the DNA sample are presented in figure 10f.

Table 2 : Global DNA hypermethylation levels obtained by SERS and standard colorimetric assay

	QOS based SERS sensor	Colorimetric assay
Fibroblast	2.730281066	2.64251
Breast Cancer	4.431938239	5.45311
Pancreatic Cancer	7.464958845	7.82156
Lung Cancer	25.58581672	26.38124

The comparison between the DNA methylation percentage obtained from QOS based SERS sensor demonstrated in this work and the colorimetric assay is presented in table 2. These results clearly show the ability of SERS based method to quantify global DNA hypermethylation levels.”

Comment: 2.2 Additionally, gene expression levels are not validated using methods like microarrays or quantitative PCR.

The authors thank the reviewer for providing this constructive comment. The authors have incorporated GAPDH as an internal control for gene expression detection. The presence of an internal standard has enabled us to calculate relative expression levels of the gene OCT-4 and SOX-2. The SERS spectra of GAPDH along with the corresponding analysis is presented in figure 11. In addition, the authors have also provided the validation data for relative gene expression in figure 12. The PCR data is incorporated in figure 12 d. Furthermore, the preciseness of tag free QOS sensor was validated using PCR and the comparative data is provided in figure 12 e,f for OCT-4 and SOX-2 respectively.

To further improve the clarity on the validation of gene expression detection the following section was added in the results in page 32 (line 8-20) :

“

Figure 12: Validation of gene expression analysis by semiquantitative PCR a, b, c) expression levels of GAPDH OCT-4 and SOX-2 in box plots respectively. D)

Semiquantitative PCR to validate relative expression levels of OCT-4 and SOX-2 in comparison with GAPDH, e, f) comparative data between relative gene expressions obtained from PCR and SERS using tag-free QOS.

The figure 12 a, b,c shows the expression levels of GAPDH OCT-4 and SOX-2 in box plots. The data is represented using the box plot as the measurements are not normally distributed. It can be observed from figure 12a, that the expression levels of GAPDH shows a similar trend in all the 4 DNA samples. Fibroblast DNA shows lower expression of both OCT-4 and SOX-2 and lung cancer DNA shows the highest expression level. PCR was performed, and the results agreed well with results from SERS detection. The PCR results in figure 12 d confirms the SERS results. In addition, the PCR results obtained are from the amplified DNA, whereas the SERS results are from the unamplified PCR product. Hence, gene expression analysis using tag-free QOS sensor provides sensitive detection on par with PCR with a higher sensitivity and reliability which is demonstrated in figure 12 e, f for relative expression levels of OCT-4 and SOX-2 respectively.”

Comment 3: The authors claim a sub-femtomolar limit of detection; however, this calculation is not done using the methods published in the literature considering signal noise, blank, and a linear slope. Also, the calibration curve only includes three data points. This plot should include a data point every decade to support log-linearity.

Response: The authors thank the reviewer for pointing out the ambiguity in the limit of detection calculation.

1. To further enhance the clarity on the calculation of limit of detection the authors have incorporated the new experimental data points for micromolar concentration of CV and R6G in figure 4 e, f for CV and figure 4 k, l for R6G.
2. In addition, to support the log-linearity of we have added new data analysis which is presented as a plot between SERS intensity vs concentration of analyte in figure 4 f, 4l for CV and R6G respectively
3. New statistical analysis was performed to determine the R value, to establish a linear relationship.

4. The R value greater than 0.9 confirming the linear relationship between intensity and analyte concentration.

The following discussion is added to the manuscript in the results section page 14,15, 17 (line 14 -20) to explain the linear relationship between SERS intensity and analyte concentration.

“

The presence of high surface area for molecular adsorption has enabled the detection of molecules in an ultra-low concentration. Figure 4e, f shows the limit of detection of CV and 4k, l shows the limit of detection of R6G. It can be observed from figure 4 e, k when the concentration

of analyte decreases, the SERS intensity decreases exhibiting a linear dependence over a wide range of concentration with a R value of 0.9895 and 0.9497 for CV and R6G respectively. It can be determined from figure 4 e, f that there exists a linear relationship between SERS intensity and analyte concentration, demonstrating the detection at single molecule level.

All the SERS spectra shows a repeatable peak positions and intensities with a relative standard deviation less than 1.6 for the characteristic peaks of CV and R6G. In addition, for the concentration of 10^{-15} M the calculated signal to noise ratio for the peak 990 cm^{-1} is 12 and for the peak 1190 cm^{-1} is 19. This confirms that QOS is a precise substrate for detection of molecules in the sub-femtomolar range.”

Comment 4: The authors discuss efficient charge transfer between the analyte and QOS as one of the two mechanisms contributing to enhancement. To support this, they present a UV/VIS spectrum; however, they do not provide the UV/VIS of the CV or RGB alone to understand the reasoning behind the spectra obtained in QOS+RGB and QOS+CV.

Response: The authors thank the reviewer for the valuable suggestion.

1. To provide further clarity on the discussion regarding charge transfer between analyte and QOS, we have conducted new experiments on the UV-visible transmission measurements of CV and R6G which is incorporated in figure 6 e, f respectively.
2. In addition, the following discussion is added to the manuscript to provide a better explanation of the mechanism.
3. From the UV-visible transmission measurements of CV and R6G, the authors were able to further confirm the presence of a probability of charge transition between analyte molecules and QOS.

The following explanation is incorporated in the manuscript in the results page 20 , 22 (line 4 -12) :

“

Figure 6: Proposed mechanism of SERS enhancement of analyte molecules adsorbed on QOS a) Charge transfer processes involved in enhancement of CV molecule b) Charge transfer processes involved in enhancement of R6G molecule c) Investigation of interaction between analyte molecules and QOS using UV-Visible transmission measurements d) Differential conductance measurements to prove the presence of inelastic phonon scattering leading to enormous SERS enhancement e)UV-visible transmission measurements of CV f) UV-visible transmission measurements of R6G

In the transmission spectra in Figure 6c, with the addition of analyte molecules there is increase in absorption, indicated by the peaks at 355 nm and 479 nm for CV and 361 nm and 484 nm for R6G. The UV-visible transmission measurements of CV presented in figure 6e shows distinct peaks at 303 nm and 471 nm and for R6G the peaks appear at 291 nm and 451 nm consistent with reported literature^[4]. These peaks are considerably red-shifted on interaction with QOS. The increase in absorption in combination with the red-shifting of the absorption peaks implies the presence of probability of electron transition between analyte molecules and QOS. From the transmission spectra, we can observe a blue shift in the absorption peak of QOS when molecular interaction happens.”

Comment 5: Not enough information is presented for the differential capacitance measurements to explain the charge transfer mechanism.

Response: The authors thank the reviewer for the valuable comments. The differential conductance measurements were incorporated to provide a better understanding of the SERS enhancement mechanism of tag free QOS sensor. The integral property of organic

semiconductors i.e. the high charge carrier density plays a predominant role in SERS enhancement. The V-shaped differential conductance curve further validates the generation of additional charge carriers, thereby enabling efficient charge transfer.

The authors have added the following information to further explain the role of differential conductance on enhancing charge transfer mechanism in the results section of the manuscript in page 22 (line 18 -28)

“The presence of a V-shaped differential conductance curve as shown in figure 5d with a dip near the fermi level suggests the presence of phonon assisted inelastic tunneling consistent with previously reported literature ^[5]. In addition, the presence of V-shaped differential conductance indicates a change in local density of states, leading to a larger charge carrier asymmetry. Further, the presence of nitrogen atoms in the carbon lattice, enhances the electron localization around the nitrogen atoms. The enhanced electron localization along with the shift in Dirac point due charge carrier asymmetry and the presence of phonon assisted inelastic tunneling is essential to generate additional charge carriers, which further transforms the charge transfer efficiency of organic semiconductors. Studies have shown that presence of heteroatoms such as nitrogen in the graphene lattice shifts the fermi energy level close to LUMO band of analyte which results in exceptional Raman enhancement”

Comment 6: Originality and significance: If the conclusions are not original, please provide relevant references. On a more subjective note, do you feel that the results presented are of immediate interest to many people in your own discipline, and/or to people from several disciplines?

The results, if properly presented and validated, are significant.

Response: The authors thank the reviewer for the valuable insight.

Based on the reviewer’s suggestion, the authors have incorporated the following new experimental data to validate the results obtained from SERS based detection using tag free QOS sensor.

1. New biological assay to validate on the global DNA hypermethylation status of genomic DNA was performed and the data is incorporated in figure 10.

2. New experimental data on creating a standard curve for DNA methylation was undertaken. The standard curve facilitated the quantification of global DNA hypermethylation in single molecule of genomic DNA.
3. New biological assay to validate gene expression detection using tag free QOS sensor was undertaken using PCR. The data analysis was incorporated in figure 12.
4. New experimental data on GAPDH- an internal standard for gene expression was performed to enable quantification of relative gene expression levels. The data is reinforced in figure 12.

Comment 7: The figures are poorly presented and are very difficult to understand. The formatting is inconsistent across multiple figures and axes titles are missing or difficult to read.

Response: The authors thank the reviewer for the valuable comment.

All the figures in the manuscript were formatted and presented in a format which is easy to understand. Further, the axis titles are changed and presented in a readable format.

Comment 8: Appropriate statistical analyses are not done to understand whether methylated and non-methylated bases are statistically significant between different bases. For example, in Figure 8b.

Response: The authors thank the reviewer for the comment.

The following explanation has been added to improve the clarity on the discussion of figure 8b.

“The global hypermethylation of cancer cell DNA can also be observed from signature peak intensity as given in Figure 9 b. The normalized peak intensities of other SERS markers of methylation including peak intensities at 837 cm^{-1} (C-C stretch) and 1120 cm^{-1} (Adenine deoxyribose) are significantly higher than that of fibroblast (figure 9b) These data show that the ratio of methylation is considerably higher in cancer cells in comparison to fibroblast cells. DNA isolated from lung cancer cells exhibit highest degree of global hypermethylation. The applicability of QOS for real time DNA methylation detection is validated and presented in figure 10.”

In addition, the necessary statistical analysis was performed and incorporated in the corresponding figures. For instance, the relative standard deviation was calculated for all the

SERS spectra presented in the manuscript. The R^2 value is reported in the respective graphs in figure 4, 10, 12. Furthermore, p values have been calculated for all the presented column graphs, only the statistically significant data were used for the corresponding analysis.

The following explanation on statistical analysis was added to the materials and methods section in the manuscript in page 35 (line 8- 18)

“To ensure the reproducibility and uniformity of signals the acquisition time was kept constant at 10s and 3 acquisitions were made. The SERS peaks used for analysis the peak intensities used for biological analysis was carefully chosen with the signal- to-noise ratio higher than 3 times the standard deviation of the data. Gene expression detection was performed using PCR assisted hybridization. The genomic DNA was only hybridized, and the unamplified sample was used for SERS detection of OCT-4 and SOX-2 expression.

All the statistical analysis were performed using XLSTAT software [6]. We used two-tailed student’s t test to compare the differences between the DNA bases, global hypermethylation levels, relative gene expression levels. We applied ANOVA (F-test) to compare the differences between the DNA isolated from various cellular models. The $P < 0.05$ was considered statistically significant in all the analysis reported in the manuscript.”

Comment 8: As mentioned earlier, lack of quantitative validation data for data presented here by SERS significantly weakens the validity of the conclusions

Response:

Based on the reviewer’s suggestion, the authors have incorporated the following new experimental data to validate the results obtained from SERS based detection using tag free QOS sensor.

1. New biological assay to validate on the global DNA hypermethylation status of genomic DNA was performed and the data is incorporated in figure 10.
2. New experimental data on creating a standard curve for DNA methylation was undertaken. The standard curve facilitated the quantification of global DNA hypermethylation in single molecule of genomic DNA.

3. New biological assay to validate gene expression detection using tag free QOS sensor was undertaken using PCR. The data analysis was incorporated in figure 12.
4. New experimental data on GAPDH- an internal standard for gene expression was performed to enable quantification of relative gene expression levels. The data is reinforced in figure 12.

Response to Reviewer 3:

The authors thank the reviewers for the encouraging comments on the novelty of the work. To further provide a better clarity on the concept and to address the reviewer's comments the following new experimental data has been added.

1. Explanation on the uniqueness of the fabricated quantum organic semiconductor SERS substrate was added to the introduction part
2. Explanation was added to improve the clarity on the mechanism of SERS enhancement by quantum organic semiconductors
3. New experimental data on the Raman spectra on bare graphite is incorporated in figure 4
4. New experimental data on SERS enhancement factors for micromolar, nanomolar concentrations of CV and R6G was incorporated in table 1.
5. New experimental data on SERS spectra of micromolar and nanomolar concentrations of CV and R6G was incorporated in Supplementary Information (Figure S1)
6. New experimental data on reproducibility of the fabricated SERS substrates was performed at 10^{-15} M concentration of the analyte concentration and incorporated in figure 5.
7. Explanation on statistical analysis carried out in the manuscript was added to the materials and methods section of the manuscript

Comment 1: In this article the authors examine the use of an organic semiconductor dot having very small size (3-7 nm) as a probe/substrate for surface-enhanced Raman spectroscopy (SERS). They synthesize carbon-dots as a SERS probe/substrate using femtosecond laser pulse irradiate on a graphite substrate in the presence of nitrogen gas. They use R6G and CV as test molecules. The authors claim up a $\sim 10^{12}$ enhancement of Raman intensities. As a real-life application, they utilize synthesized organic semiconductor dots in the detection of cancer stem cell marker OCT4 and SOX2. Although graphene and graphene-based quantum dots have been utilized for many years (especially for the last 3 years) as SERS substrate, this is the first use, to my knowledge, of a graphene-based quantum dot in the detection of CSC markers. It is here that this paper is interesting.

Response: The authors thank the reviewer for the highlighting the novelty of the manuscript.

Comment 2: Unfortunately, the authors seem quite unaware of the existing experiments (Nat. Commun. 2018, 9, 193; Nanoscale 2016, 8, 8863 etc.). Therefore, the SERS substrates they fabricated are not novel.

Response: The authors would like to thank the reviewer for the valuable comment. The authors would like to further clarify the novelty of the using femtosecond laser synthesis of quantum organic semiconductor SERS substrates:

1. Although, there have been previous studies on femtosecond laser-based fabrication on carbon -based quantum scale materials like graphene quantum dots and carbon quantum dots. To the best of our knowledge, there exists no attempt to synthesize quantum scale organic semiconductors.
2. Femtosecond laser synthesis of quantum dots involves the precise manipulation of ionization, condensation and atomic recombination of ionic species present in the ablated plume. The traditional femtosecond laser ablation uses liquid medium for condensation and recombination process. However, due to inherent difficulties present in the synthesis of quantum dots in liquid environment, it cannot be applied to synthesize quantum organic semiconductor.
3. Furthermore, the earlier attempts were made by using femtosecond laser for synthesis of graphene quantum dots in colloidal state. The quantum dots in colloidal state have proven to be inept SERS substrates owing to extensive particle aggregation which in turn leads to inconsistent spectral signature.
4. Recently, Yilmaz et.al has pioneered the way to use organic semiconductor as an efficient SERS substrate. The enhancement efficiency of organic semiconductor in this work was low due to the use of thin film, where the primary SERS enhancement mechanism is charge transfer. Although organic semiconductor possesses extraordinary properties, the low enhancement efficiency impedes the ability to use for ultrasensitive detection purposes.
5. In this work, the authors use reduced pulse to pulse separation time and pulse energy to precisely control the process of ionization. Additionally, the use of low-pressure nitrogen (dry process) as a medium to assist condensation and atomic recombination results in the shrinking of organic semiconductor to quantum scale. The low-pressure nitrogen

environment aids in incorporating nitrogen atom in the carbon lattice which is essential to determine the properties of organic semiconductors.

6. The ultrasensitive detection of epigenetic contribution of CSC using Single molecule of genomic DNA was made possible by shrinking organic semiconductor to quantum size.
7. As the reviewer suggested, the following references on fabrication of graphene quantum dots using femtosecond laser, was added to bring more clarity on the novelty of the fabricated QOS

- [7] J. Perrière, C. Boulmer-Leborgne, R. Benzerga, S. Tricot, *J. Phys. D. Appl. Phys.* **2007**, *40*, 7069.
- [8] C. H. Nee, S. L. Yap, T. Y. Tou, H. C. Chang, S. S. Yap, *Sci. Rep.* **2016**, *6*, 1.
- [9] N. Sudani, K. Venkatakrishnan, B. Tan, *Mater. Manuf. Process.* **2011**, *26*, 661.

To provide a clear understanding regarding the novelty of the fabricated SERS substrate, the following section is added to the manuscript in the introduction:

“Currently, no synthesis method exists for quantum scale organic semiconductors has been reported. The shrinking of organic semiconductor to quantum scale is realizable only by a precise manipulation of ionization and condensation process in a solid substrate using femtosecond laser. Femtosecond laser synthesis of quantum dots involves the precise manipulation of ionization, condensation and atomic recombination of ionic species present in the ablated plume. Earlier attempts to synthesize graphene quantum dot synthesis using femtosecond laser ablation carbon based material were reported [7-9]. The traditional femtosecond laser ablation uses liquid medium for condensation and recombination process. However, due to inherent difficulties present in the synthesis of quantum dots in liquid environment, it cannot be applied to synthesize quantum organic semiconductor. These studies use femtosecond laser to synthesize particles in colloidal state, which are proven to be inefficient SERS substrates owing to extensive particle aggregation which in turn leads to inconsistent spectral signature [10]. To the best of our knowledge, there is no attempt to synthesize quantum scale organic semiconductors.”

Comment 2: The authors propose a new SERS enhancement mechanism called electron mobility enhanced Raman. However, their discussion is mainly based on speculation (there is no

experimental data to support their claims). They also claim that the record enhancement factor of 10^{12} is due to the electromagnetic enhancement which is originated from the surface plasmons of organic semiconductor dots. Again, there is no experimental proof to support this point.

Response: The authors thank the reviewer for the valuable observation. The authors would like to provide an improved explanation for the mechanism of SERS enhancement in the QOS sensor.

1. The SERS enhancement observed in the QOS system is an interplay between surface plasmon, charge-transfer and molecular resonance.
2. In the case of quantum organic semiconductors, the surface plasmon is highly influenced by charge carrier density^[11] which was recently observed by Agarwal et.al.
3. In addition, the presence of heteroatoms such as nitrogen, oxygen in the carbon lattice have proven to induce localized surface plasmons in semiconductor nanostructures with free charge carriers^[12].
4. The role of enhanced charge carriers on SERS enhancement is explained using the differential conductance curve shown in figure 6d. A clear explanation on the role of charge carrier density in SERS enhancement of QOS is added to the manuscript.

The following section is added to the result in page 22 (line 18 -28)

“The presence of a V-shaped differential conductance curve as shown in figure 6d with a dip near the fermi level suggests the presence of phonon assisted inelastic tunneling consistent with previously reported literature^[5]. In addition, the presence of V-shaped differential conductance indicates a change in local density of states, leading to a larger charge carrier asymmetry. Further, the presence of nitrogen atoms in the carbon lattice, enhances the electron localization around the nitrogen atoms. The enhanced electron localization along with the shift in Dirac point due charge carrier asymmetry and the presence of phonon assisted inelastic tunneling is essential to generate additional charge carriers, which further transforms the charge transfer efficiency of organic semiconductors. Studies have shown that presence of heteroatoms such as nitrogen in the graphene lattice shifts the fermi energy level close to LUMO band of analyte which results in exceptional Raman enhancement.”

Comment 3: For the EF calculations, the authors use picomolar (10^{-12} M) concentrations of analyte molecules. I think that's why they observed such a high EF. If they used conventional concentrations such as 10^{-3} or 10^{-5} M, they would get much lower EF.

Response: The author would like to thank the reviewer for the valuable suggestion.

1. The authors have added enhancement factor calculations for 2 more concentrations (micromolar, nanomolar) in addition to picomolar, femtomolar as suggested by reviewer.
2. The new SERS spectra of micromolar, nanomolar concentrations of CV and R6G was incorporated in supplementary information in figure S1
3. Based on the calculations, we observe a decreasing trend in the enhancement factor with increasing number of molecules.
4. In addition, the authors have also added new table 1, containing the enhancement factors for the various analyte concentration in the results section page 18.

Table 1: SERS enhancement factors of Cv and R6G at different concentrations

Molar Concentration	CV	R6G
10^{-6}	3.91×10^{10}	4.85×10^{10}
10^{-9}	3.23×10^{10}	4.56×10^{10}
10^{-12}	1.11×10^{12}	4.57×10^{12}
10^{-15}	1.39×10^{12}	4.43×10^{12}

Figure S1: a) SERS spectra of 10^{-6} M and 10^{-9} M concentration of CV on QOS b) SERS spectra of 10^{-6} M and 10^{-9} M concentration of R6G on QOS

Comment 4: - In fig4, there is no Raman spectra of analytes on bare graphite. Moreover, there are some shifting in peak positions unlike authors claim.

Response: The authors thank the reviewer for the valuable comment.

The authors have added the Raman spectra for graphite in figure 4 for both the small QOS and Large QOS and the same is incorporated in figure 4. Additionally, the peak shift observed are within the standard peak shift present based on the allowed standard error margin for each instrument.

Comment 5: - I do not understand the sentence in page 15 “The electronegative oxygen functional groups in the 3D architecture creates a local electric field on the analyte molecules, thus enabling a hotspot formation upon laser excitation”. How oxygen atoms create electromagnetic hot spots?

Response: The authors thank the reviewer for the valuable observation. The authors would like to explain the sentence as follows

The XPS results presented in figure 2 indicates the presence of oxygen functional groups in the QOS. Recent literature has confirmed that of oxygen functional groups tends to be on the surface of quantum dots. In addition, the oxygen functional groups often form a cluster like islands on the surface, which acts like hotspots. Also, when the analyte is adsorbed on the surface of QOS, the oxygenated functional group along with the local defects leads to a highly enhanced Raman signal.

The following section was added to the results section in the manuscript to enhance the clarity

“The electronegative oxygen functional groups in the 3D architecture creates a local electric field on the analyte molecules, thus mimicking a hotspot formation upon laser excitation^[13]. The XPS results presented in figure 2 indicates the presence of oxygen functional groups in the QOS. Recent literature has confirmed that of oxygen functional groups tends to be on the surface of quantum dots. In addition, the oxygen functional groups often form a cluster like islands on the surface, which acts like hotspots. Also, when the analyte is adsorbed on the surface of QOS, the oxygenated functional group along with the local defects leads to a highly enhanced Raman signal^[14].”

Comment 6: - In Fig 6, 7, 8 and 9, the signal to noise ratio seems very low and it is hard to use these Raman spectra in the reliable detection of biological molecules.

Response: The authors thank the reviewer for the valuable suggestions. To address the reviewer’s concerns,

1. New experimental data was collected for figure 6,7,8,9 (original manuscript) and is incorporated in figure 7,8,9,11 in the revised manuscript.

2. In addition, the peak intensities used for biological analysis was carefully chosen with the signal- to-noise ratio higher than 3 times the standard deviation of the data.
3. Furthermore, the biological analysis reported in the manuscript was validated with standard assays.

The following figures were modified in the manuscript

Figure 7: Single molecule analysis of genomic DNA to determine base composition and DNA methylation from a single SERS measurement

Figure 8: Base composition analyses of genomic DNA a) Schematic representation of base composition analyses using SERS b) Normalized SERS peak intensity for various nucleic acid bases in genomic DNA of fibroblast, Breast Cancer, Pancreatic Cancer and Lung Cancer cells, c) PC loading to determine the differences in the genomic DNA of the cellular models aiding in cancer detection d) PCA analysis of genomic DNA derived from fibroblast, Breast Cancer, Pancreatic Cancer and Lung Cancer cells

Figure 11: Gene expression detection using QOS a, c, e) SERS spectra of GAPDH, OCT-4 and SOX-2 respectively, b, d, f) expression ratio of GAPDH, OCT-4 and SOX-2 respectively,

The following explanation was added to the materials and methods to provide further clarity on the statistical analysis carried out in the manuscript

“All the statistical analysis were performed using XLSTAT software ^[6]. We used two-tailed student’s t test to compare the differences between the DNA bases, global hypermethylation levels, relative gene expression levels. We applied ANOVA (F-test) to compare the differences between the DNA isolated from various cellular models. The $P < 0.05$ was considered statistically significant in all the analysis reported in the manuscript”

Comment 7: There is also no data about the reproducibility of the fabricated substrates/probes in SERS which is important for real life applications.

Response: The authors thank the reviewer for the constructive comments.

1. New experiment was performed for CV and R6G at 10^{-15} M concentration and the data was incorporated in figure 5 in the revised manuscript.
2. The reproducibility of the fabricated SERS substrates was added as a separate section in the manuscript.
3. Further, detailed analysis was provided. Based on the data obtained, it can be confirmed that QOS is a reliable substrate for ultrasensitive SERS detection.

The following changes has been incorporated in the manuscript to address the comment on reproducibility:

Repeatability and Reproducibility of SERS enhancement:

Figure 5: Reproducibility analysis of SERS signals of a, b) CV, R6G on QOS respectively on QOS, c) intensities of characteristic CV peak at 1613 cm^{-1} shown in spectra (a) error bars: SD, d) intensities of characteristic R6G peak at 1643 cm^{-1} shown in spectra (b) error bars: SD.

In the traditional metal-based SERS system, repeatability and reproducibility of SERS signals are the major bottlenecks. Hence, we tested the reproducibility of the obtained SERS signals by measuring the SERS spectra at 5 different points. The obtained spectra are shown in figure 5 a, b for CV and R6G respectively. It can be observed from figure 5a showing consistent reproducible characteristic SERS peak of CV at 1613 cm^{-1} . The corresponding peak intensity shown in figure 5c at 5 different points show a little difference with a relative standard deviation of 1.68% , thus confirming excellent reproducibility. The similar results for R6G is shown in figure 5b. The spectra showcase a consistent peak at 1643 cm^{-1} and corresponding uniform peak intensity with

an RSD of 2.92% is shown in figure 5d. The presence of uniform intensity with a low RSD value indicates the reliability of nanostructured QOS as a SERS substrate”

The following references have also been incorporated in the manuscript:

References:

- [1] P. Lin, F. Yan, Organic thin-film transistors for chemical and biological sensing. *Adv. Mater.* **2012**.
- [2] T. H. Kim, D. Lee, J. W. Choi, *Biosens. Bioelectron.* **2017**, *94*, 485.
- [3] S. Cong, Y. Yuan, Z. Chen, J. Hou, M. Yang, Y. Su, Y. Zhang, L. Li, Q. Li, F. Geng, Z. Zhao, *Nat. Commun.* **2015**, *6*, 1.
- [4] S. Feng, M. Cristina dos Santos, B. R. Carvalho, R. Lv, Q. Li, K. Fujisawa, A. L. Elías, Y. Lei, N. Perea-López, M. Endo, M. Pan, M. A. Pimenta, M. Terrones, *Sci. Adv.* **2016**, *2*, 1.
- [5] R. Lv, Q. Li, A. R. Botello-Méndez, T. Hayashi, B. Wang, A. Berkdemir, Q. Hao, A. L. Eléas, R. Cruz-Silva, H. R. Gutiérrez, Y. A. Kim, H. Muramatsu, J. Zhu, M. Endo, H. Terrones, J. C. Charlier, M. Pan, M. Terrones, *Sci. Rep.* **2012**, *2*, 1.
- [6] Addinsoft (2019). XLSTAT statistical and data analysis solution.
- [7] J. Perrière, C. Boulmer-Leborgne, R. Benzerga, S. Tricot, *J. Phys. D. Appl. Phys.* **2007**, *40*, 7069.
- [8] C. H. Nee, S. L. Yap, T. Y. Tou, H. C. Chang, S. S. Yap, *Sci. Rep.* **2016**, *6*, 1.
- [9] N. Sudani, K. Venkatakrishnan, B. Tan, *Mater. Manuf. Process.* **2011**, *26*, 661.
- [10] S. S. Sinha, S. Jones, A. Pramanik, P. C. Ray, *Acc. Chem. Res.* **2016**, *49*, 2725.
- [11] A. Agrawal, S. H. Cho, O. Zandi, S. Ghosh, R. W. Johns, D. J. Milliron, *Chem. Rev.* **2018**, *118*, 3121.
- [12] J. M. Luther, P. K. Jain, T. Ewers, A. P. Alivisatos, *Nat. Mater.* **2011**, *10*, 361.
- [13] H. Xu, L. Yan, V. Nguyen, Y. Yu, Y. Xu, *Appl. Surf. Sci.* **2017**, *414*, 238.
- [14] X. Yu, H. Cai, W. Zhang, X. Li, N. Pan, Y. Luo, X. Wang, J. G. Hou, *ACS Nano* **2011**, *5*, 952.

Reviewers' comments:

Reviewer #1 (Remarks to the Author):

The authors have adequately taken into account my suggestions and I recommend publication.

Reviewer #2 (Remarks to the Author):

My main concern with the manuscript was that control and validation studies were not adequately performed. With the new sets of experiments, the authors have addressed this issue in a satisfactory manner.

Reviewer #3 (Remarks to the Author):

The authors deeply revised the whole manuscript in light of the reviewers' comments and now the work looks more clear. They have done a lot of additional work.

However my opinion didn't change that much. About the EF calculations, I have still doubts (if authors can provide more detailed calculations by giving analyte spot size, number of analyte in laser spot, Raman intensity etc., it could be more convincing). Because authors claim that the main Raman enhancement mechanism is based on the charge-transfer and/or charge-transfer resonances, the large EF value as 10^{10} is not realistic (There is still no data about charge-transfer (PL or pump-probe spec. data should be provided to show clear charge transfers between substrate and analyte). They also claim that electromagnetic enhancement is contributed to over all enhancement, but still there is no data about it in the manuscript (I could not find any plasmonic evidence resonating with 785 nm of Raman laser excitation).

From the material perspective (organic semiconductor dots), this work still suffers lack of novelty and cannot be considered at this level of importance (the one asked for Nature Communications).

In conclusion I am sorry to say that I cannot recommend publication in Nature Communications.

Point-by -point response to reviewers

Reviewer #1

Comment 1: The authors have adequately taken into account my suggestions and I recommend publication.

Response: The authors thank the reviewer for the constructive review process.

Reviewer #2

Comment 1: My main concern with the manuscript was that control and validation studies were not adequately performed. With the new sets of experiments, the authors have addressed this issue in a satisfactory manner.

Response: The authors thank the reviewer for productive review process

Reviewer #3

Comment 1: The authors deeply revised the whole manuscript in light of the reviewers' comments and now the work looks more clear. They have done a lot of additional work.

Response: The authors thank the reviewer for the positive comment.

To further address the reviewer's concern regarding the enhancement factor calculations and the mechanism of charge transfer enhancement, we have added the following additional experimentation and explanation to the manuscript.

Comment 2: About the EF calculations, I have still doubts (if authors can provide more detailed calculations by giving analyte spot size, number of analyte in laser spot, Raman intensity etc., it could be more convincing).

Response: The authors thank the reviewer for the comment. To provide further clarity on the enhancement factor calculation, the authors have added a new section to the materials and method section, providing a detailed explanation on the method followed to calculate the enhancement factor as reported in literature. Further, as requested by the reviewer, we have provided the calculation and the values used for arriving at the enhancement factor as a separate table in the materials and methods section of the manuscript.

The following explanation is added to the materials and methods section:

“The enhancement factor was calculated according to the calculations provided in the literature

¹⁻⁴. The enhancement factor per molecule is calculated using the equation:

$$EF = \frac{\frac{I_{QOS}}{N_{QOS}}}{\frac{I_{graphite}}{N_{graphite}}}$$

$$N_{graphite} = \pi r^2 h c N_A$$

$$N_{QOS} = SA_{eff} \left(C_{ads} N_A \frac{1000 L}{1 m^3} \right)^{2/3}$$

Where I is the intensity of the peak selected, N is the number of molecules in the laser spot. The laser spot size used for calculating EF is 1 μm (0.5 μm radius), the laser penetration depth was measured as 10 μm based on 20X long working distance objective. Hence, the volume in the laser spot size 26.17 μm^3 . Then by using the values for crystal violet (density 0.981 g/cm^3 , molar mass 407.979 g/mol) and Rhodamine 6G (density 1.26 g/cm^3 , molar mass 479.02 g/mol) to find the number of molecules in the laser spot size. Using the above-mentioned values, the values of $N_{graphite}$ as 1.047×10^{16} , 1.579×10^{16} for crystal violet and rhodamine 6G, respectively. The value of N_{QOS} was obtained by dividing the laser surface area (6.28 μm^2) by the effective cross-sectional area per molecule (6×10^{-7})^{3,5,6}. Hence, the calculated value of N_{QOS} to obtain the number of molecules on QOS 1.04×10^7 . The values of I_{QOS} and $I_{graphite}$ was obtained from the SERS spectra. the calculation for picomolar concentration of analyte is provided in the table below:

	CV	R6G
Spot volume (μm^3)	26.17	26.17
Density ($\text{g}/\mu\text{m}^3$)	9.81E+11	1.26E+12
molar mass (g/mol)	407.979	479.02
$N_{graphite}$	1.0474E+16	1.57953E+16
SA of spot (μm^2)	6.28	6.28
effective cross section	0.0000006	0.0000006
N_{QOS}	10466666.67	10466666.67
$N_{graphite} / N_{QOS}$	1.635E+18	1509105218
$I_{QOS}/I_{graphite}$	1106.048311	3026.804331
EF	1.80839E+12	4.56777E+12

The following references were added to support the method of calculation of enhancement factor:

1. Cong, S. *et al.* Noble metal-comparable SERS enhancement from semiconducting metal oxides by making oxygen vacancies. *Nat. Commun.* **6**, 1–7 (2015).
2. Ling, X. *et al.* Raman enhancement effect on two-dimensional layered materials: Graphene, h-BN and MoS2. *Nano Lett.* **14**, 3033–3040 (2014).

- Muehlethaler, C. *et al.* Ultrahigh Raman Enhancement on Monolayer MoS₂. *ACS Photonics* **3**, 1164–1169 (2016).
- Zheng, Z. *et al.* Semiconductor SERS enhancement enabled by oxygen incorporation. *Nat. Commun.* (2017). doi:10.1038/s41467-017-02166-z
- Soriaga, M. P. & Hubbard, A. T. Determination of the Orientation of Adsorbed Molecules at Solid-Liquid Interfaces by Thin-Layer Electrochemistry: Aromatic Compounds at Platinum Electrodes. *J. Am. Chem. Soc.* **104**, 2735–2742 (1982).

Comment 3.1: Because authors claim that the main Raman enhancement mechanism is based on the charge-transfer and/or charge-transfer resonances, the large EF value as 10^{10} is not realistic (There is still no data about charge-transfer (PL or pump-probe spec. data should be provided to show clear charge transfers between substrate and analyte)).

Response: The authors thank the reviewer for the valuable comment. The authors have added new experimentation to provide a clear understanding about the charge transfer transitions happening between substrate and analyte. As suggested by the reviewer, we have utilized PL spectroscopy to explain in detail the charge transfer transitions between the substrate and analyte.

The following explanation and figure 6 were added to the manuscript in page 20 – 22

“

The relationship between molecular adsorption and charge transfer between analyte and QOS, was studied using photoluminescence spectra presented in figure 6 g, h for CV and R6G respectively. On observing the spectra, it should be noted that the PL intensity significantly drops on interaction with QOS. This trend is observed in both CV and R6G, although the decrease in intensity for R6G adsorption is lower. This minor intensity drop suggests that the presence of resonant charge transfer in R6G and a non-resonant charge transfer in CV⁷. Further, comparing the PL intensity of the molecule and molecule/ QOS complex, it can be observed that the molecule/QOS complex exhibit decreased PL intensity. The decreased PL intensity implies the

presence of exciton quenching happening as a result of electron transfer between molecule and QOS⁸. The other evidence of charge transfer observed from the PL spectra in figure 6 g, h is the blue shift of CV peak by 25 meV and R6G by 65 meV. In addition, we also observe a narrowing of the peak in both CV (FWHM = 51) CV + QOS (FWHM = 47) and for R6G (FWHM = 85) R6G +QOS (FWHM = 54). The blue shift of PL along with the narrowing of peaks, may be attributed to alterations in exciton recombination leading to efficient charge transfer^{9,10}. On establishing the presence of charge transfer between analyte and QOS, further explanation on the possible charge transfer mechanism is discussed below.”

The following references were added to prove the direct observation of charge transfer between QOS and analyte molecule:

6. Villaeys, A. A. & Zouari, M. Role of the substrate field inhomogeneities in coherent resonant raman scattering. *J. Phys. Chem. A* **111**, 9522–9531 (2007).
7. Katayama, K., Shibamoto, K. & Sawada, T. Direct observation of ultrafast charge transfer in relation to the surface enhanced Raman scattering activation detected by transient reflecting grating spectroscopy. *Chem. Phys. Lett.* **345**, 265–271 (2001).
8. Garcia-Basabe, Y. *et al.* Ultrafast charge transfer dynamics pathways in two-dimensional MoS₂-graphene heterostructures: A core-hole clock approach. *Phys. Chem. Chem. Phys.* **19**, 29954–29962 (2017).
9. Buscema, M., Steele, G. A., van der Zant, H. S. J. & Castellanos-Gomez, A. The effect of the substrate on the Raman and photoluminescence emission of single-layer MoS₂. *Nano Res.* **7**, 1–11 (2014).
10. Ago, H. *et al.* Controlled van der Waals epitaxy of monolayer MoS₂ triangular domains on graphene. *ACS Appl. Mater. Interfaces* **7**, 5265–5273 (2015).

Comment 3.2: They also claim that electromagnetic enhancement is contributed to over all enhancement, but still there is no data about it in the manuscript (I could not find any plasmonic evidence resonating with 785 nm of Raman laser excitation).

Response: The authors have claimed the mechanism of SERS enhancement is due a combined effect of surface plasmon resonance, charge transfer and quantum confinement effect. To provide more clarity on the enchantment mechanism and to assert the presence of surface plasmon resonant with the 785 nm Raman excitation wavelength, we have used LSPR spectroscopy to study the λ max for QOS. The presence of LSPR at 725 nm validates that 785 nm Raman excitation wavelength is resonant with QOS, thereby enhancing SERS efficiency. The following figure and explanation were added to the manuscript to provide more clarity on the presence of surface plasmon.

“The plasmon resonance of semiconductors is highly dependent on the charge density. Generally, the plasmon resonance of organic semiconductors lies in infrared region, hence the excitation wavelength of 785nm has a great influence on the enhancement efficiency of QOS. The LSPR spectra of QOS is shown in figure 6 i. It can be inferred from the figure 6i that the LSPR of QOS is present at 725 nm, thus validating the utilization of 785nm as the Raman excitation wavelength. Further, the presence of nitrogen atoms in the carbon lattice helps in improving the plasmon propagation length, thereby leading to a high SERS enhancement¹¹⁻¹³.”

The following references were added to support the presence of resonant surface plasmon

11. Jablan, M., Buljan, H. & Soljačić, M. Plasmonics in graphene at infrared frequencies. *Phys. Rev. B - Condens. Matter Mater. Phys.* **80**, 1–7 (2009).
12. Koppens, F. H. L., Chang, D. E. & García De Abajo, F. J. Graphene plasmonics: A platform for strong light-matter interactions. *Nano Lett.* (2011). doi:10.1021/nl201771h
13. Novko, D. Dopant-induced plasmon decay in graphene. 1–17

Reviewers' comments:

Reviewer #3 (Remarks to the Author):

I would like to thank all the authors for their kind response to my comments. However, my opinion still didn't change. The estimation of the number of molecules contributing to the SERS enhancement on QOS is not reliable (and also if I understood the authors' response correctly, the number of molecules on graphite is also wrong (please see below)). To do this estimation, several important factors need to be considered (please see ACS Photonics 2016, 3, 1164). According to the authors' estimation, the number of molecules on QOS is about 10^7 . But they said that this calculation is for picomolar concentration of analyte (10^{-12} mol/L). In this concentration, there are about 10^{11} molecules in 1 L of solution. Therefore, the estimation seems to be revised or changed.

In supporting Figure S1a, why 10⁻⁶ M CV has lower Raman intensity than 10⁻⁹ M CV concentration? In Figure S1b, the Raman spectrum of 10⁻⁶ M R6G is obviously different than the Raman spectrum of 10⁻⁹ M R6G. Why?

Finally, How the presence of LSPR at 725 nm validates that 785 nm Raman excitation wavelength is resonant with QOS?

In conclusion I am sorry to say that I still cannot recommend this manuscript for the publication in Nature Communications.

$$\text{Laser spot area} = \pi \times r^2 = 3.14 \times (0.5 \mu\text{m})^2 = 0.785 \mu\text{m}^2$$

$$\text{Optical Excitation volume (Vopt)} = \pi \times r^2 \times h = 0.785 \mu\text{m}^2 \times 10 \mu\text{m} = 7.85 \mu\text{m}^3$$

$$\text{Ngraphite for R6G} = ((V_{\text{opt}} \times \text{Density})/M) \times N_A = ((7.85 \times 1.26 \times 10^{-12})/479.02) \times 6.02 \times 10^{23} = 1.24 \times 10^{12} \text{ R6G}$$

Response to reviewers

With reference to the letter regarding revision of the manuscript titled “**Shrinking Organic Semiconductors to Quantum Scale – Utilizing Genomic DNA for Cancer Stem Cell Detection - NCOMMS-19-10288B**”. The authors would like to thank the reviewers for the constructive review process. However, we believe any further revision is unlikely to yield a meaningful improvement of the work.

To address the questions raised by reviewer #3, the explanation is provided below:

Query 1: I would like to thank all the authors for their kind response to my comments. However, my opinion still didn't change. The estimation of the number of molecules contributing to the SERS enhancement on QOS is not reliable (and also if I understood the authors' response correctly, the number of molecules on graphite is also wrong (please see below)). To do this estimation, several important factors need to be considered (please see ACS Photonics 2016, 3, 1164).

Response to query 1: The number of molecules contributing to SERS enhancement was calculated according to the enhancement factor calculation commonly followed by leading researcher's in the field of SERS. We consider the enhancement factor calculation commonly followed in various highly cited works published in journals including Nature Communications is best suited for our work. The estimations and assumptions essential for calculating the enhancement factor were adapted from the paper ACS Photonics 2016, 3, 1164 mentioned by the reviewer, which is already cited as reference 39 in the initial submission.

Moreover, the enhancement factor calculation provided by the reviewer is an alternative approach which is followed only by a limited number of researchers. Furthermore, the calculation provided by the reviewer for determining the number of molecules assumes the optical excitation volume to be cylindrical ($\pi r^2 h$) whereas the actual optical excitation spot is considered hemispherical ($2/3 \pi r^3 * h * \text{Numerical Aperture}$). Hence, we believe the alternative calculation of enhancement factor will not be suitable for our work.

However, in order to provide more clarity regarding the enhancement factor we can add both the enhancement factor calculations in the Supporting information of the manuscript.

The following references support the validity and reliability of the enhancement factor calculations used in this work:

1. Muehlethaler, C. et al. Ultrahigh Raman Enhancement on Monolayer MoS₂. ACS Photonics 3, 1164–1169 (2016).
2. Lombardi, J. R. & Birke, R. L. Theory of surface-enhanced raman scattering in semiconductors. J. Phys. Chem. C 118, 11120–11130 (2014).
3. Zheng, Z. et al. Semiconductor SERS enhancement enabled by oxygen incorporation. Nat. Commun. (2017). doi:10.1038/s41467-017-02166-z
4. Cong, S. et al. Noble metal-comparable SERS enhancement from semiconducting metal oxides by making oxygen vacancies. Nat. Commun. 6, 1–7 (2015).

Query 2 : According to the authors' estimation, the number of molecules on QOS is about 10^7 . But they said that this calculation is for picomolar concentration of analyte (10^{-12} mol/L). In this concentration, there are about 10^{11} molecules in 1 L of solution. Therefore, the estimation seems to be revised or changed.

Response query 2: We would also like to clarify the reviewer's assumption regarding the calculation of the number of molecules. Although, the number of molecules in a picomolar concentration of solution is $10^{11}/L$, the volume used to obtain SERS spectra is 10 μ L. Moreover, only the molecules present in the laser spot size contributes to SERS signals. Hence, the number of molecules present in the laser spot size (10^7) is only used for calculation of enhancement factor.

Query 3 : In supporting Figure S1a, why 10^{-6} M CV has lower Raman intensity than 10^{-9} M CV concentration? In Figure S1b, the Raman spectrum of 10^{-6} M R6G is obviously different than the Raman spectrum of 10^{-9} M R6G. Why?

Response query 1: It is well-known that the enhancement of molecular vibrations is highly dependent on the orientation of the molecule on the SERS probe. Moreover, all the molecular vibrations are not enhanced at a similar level. However, the characteristic vibration of CV has a higher intensity in 10^{-6} M compared to 10^{-9} M. In addition, the selective enhancement at lower concentrations and the peak shifts observed in R6G can be attributed to several factors including

- When analyte concentration decreases, the overall fluorescence background also reduces, leading to prominent SERS bands.

- Additionally, when the concentration of analyte decreases, the molecules tend to be closer to the quantum probes, in which case, the SERS intensity as well as the sensitivity to molecular vibrations increases drastically.

The above-mentioned factors are confirmed by earlier published research where a similar phenomenon was observed at higher analyte concentrations.^{5,6}

Kneipp, K., Kneipp, H. & Kneipp, J. Surface-enhanced raman scattering in local optical fields of silver and gold nanoaggregates - From single-molecule raman spectroscopy to ultrasensitive probing in live cells. *Acc. Chem. Res.* 39, 443–450 (2006).

Szekeres, G. P. & Kneipp, J. SERS Probing of Proteins in Gold Nanoparticle Agglomerates. *Front. Chem.* 7, 1–10 (2019).

The detailed explanation regarding the selective enhancement of peaks is already included in the previous revisions of the manuscript at different instances as follows :

In page 16

“The presence of high surface area for molecular adsorption has enabled the detection of molecules in an ultra-low concentration. Figure 4e, f shows the limit of detection of CV and 4k, l shows the limit of detection of R6G. It can be observed from figure 4 e, k when the concentration of analyte decreases, the SERS intensity decreases exhibiting a linear dependence over a wide range of concentration with a R value of 0.9895 and 0.9497 for CV and R6G respectively. It can be determined from figure 4 e, f that there exists a linear relationship between SERS intensity and analyte concentration, demonstrating the detection at single molecule level. The SERS spectra of 10^{-6} M and 10^{-9} M concentration of CV and R6G is presented in figure S1. It should be noted that the characteristic peaks of CV and R6G selectively enhanced at higher concentrations. This could be attributed to various factors including molecular orientation, reduction in overall fluorescence background, particle aggregation at higher concentration leading to lower intensity of certain molecular vibrations⁴³.”

In page 20

“The SERS spectra in Figure 4 a b shows an intense peak corresponding to 612 cm^{-1} and Figure 4 e f shows a peak at 778 cm^{-1} . These peaks correspond to in-plane and out-of-plane bending motion, consistent with charge transfer through vibronic coupling¹⁴. The decrease in the

intensity of these peaks with decreasing concentration indicate that charge transfer is strongly dependent on molecular adsorption and orientation of the molecule. The relative low intensity at femtomolar concentration indicates that the aromatic rings in CV and R6G were not parallel to QOS surface¹⁴.”

Query 4 : Finally, How the presence of LSPR at 725 nm validates that 785 nm Raman excitation wavelength is resonant with QOS?

Response query 4: The LSPR spectra confirms the presence of surface plasmon in the quantum organic semiconductor probes. It is a universally accepted fact in the field of SERS, that the presence of LSPR peak near the Raman excitation wavelength is enough for resonance excitation. For instance, the work by (Witlicki et.al 2010, JACS) on gold nanoparticles reported a LSPR peak at 765 nm claims the presence of Resonant Raman scattering with a Raman excitation wavelength of 785 nm.

In addition, research previously published on traditional SERS materials including metal nanoparticles and semiconductors have reported an LSPR peak near the Raman excitation wavelength and not exactly at the excitation wavelength.

The following section was added during the earlier revisions in page 20 to explain the same:

“The plasmon resonance of semiconductors is highly dependent on the charge density. Generally, the plasmon resonance of organic semiconductors lies in infrared region, hence the excitation wavelength of 785nm has a great influence on the enhancement efficiency of QOS. The LSPR spectra of QOS is shown in figure 6 i. It can be inferred from the figure 6i that the LSPR of QOS is present at 725 nm, thus validating the utilization of 785nm as the Raman excitation wavelength. Further, the presence of nitrogen atoms in the carbon lattice helps in improving the plasmon propagation length, thereby leading to a high SERS enhancement⁴⁷⁻⁴⁹.”

Further, the following are some of the highly cited works to corroborate the point:

Boerigter, C., Campana, R., Morabito, M. & Linic, S. Evidence and implications of direct charge excitation as the dominant mechanism in plasmon-mediated photocatalysis. *Nat. Commun.* 7, 1–9 (2016).

Agrawal, A. et al. Localized Surface Plasmon Resonance in Semiconductor Nanocrystals. *Chem. Rev.* 118, 3121–3207 (2018).

Hossain, M. K., Kitahama, Y., Huang, G. G., Han, X. & Ozaki, Y. Surface-enhanced raman scattering: Realization of localized surface plasmon resonance using unique substrates and methods. *Anal. Bioanal. Chem.* 394, 1747–1760 (2009).

Faulds, K., Smith, W. E. & Graham, D. Evaluation of Surface-Enhanced Resonance Raman Scattering for Quantitative DNA Analysis. *Anal. Chem.* 76, 412–417 (2004).

REVIEWERS' COMMENTS:

Reviewer #3 (Remarks to the Author):

The current version of the manuscript addresses all my comments. It is more focused and includes additional analysis. The authors improved the manuscript considerably in all aspects. I, therefore, recommend accepting it with no further revision.

Response to Reviewer

The current version of the manuscript addresses all my comments. It is more focused and includes additional analysis. The authors improved the manuscript considerably in all aspects. I, therefore, recommend accepting it with no further revision.

The authors thank the reviewer for accepting the manuscript.